# Tractable Regularization of Probabilistic Circuits

**Anji Liu**
Department of Computer Science
UCLA
Los Angeles, CA 90095
liuanji@cs.ucla.edu

**Guy Van den Broeck**
Department of Computer Science
UCLA
Los Angeles, CA 90095
guyvdb@cs.ucla.edu

## Abstract

Probabilistic Circuits (PCs) are a promising avenue for probabilistic modeling. They combine advantages of probabilistic graphical models (PGMs) with those of neural networks (NNs). Crucially, however, they are tractable probabilistic models, supporting efficient and exact computation of many probabilistic inference queries, such as marginals and MAP. Further, since PCs are structured computation graphs, they can take advantage of deep-learning-style parameter updates, which greatly improves their scalability. However, this innovation also makes PCs prone to overfitting, which has been observed in many standard benchmarks. Despite the existence of abundant regularization techniques for both PGMs and NNs, they are not effective enough when applied to PCs. Instead, we re-think regularization for PCs and propose two intuitive techniques, *data softening* and *entropy regularization*, that both take advantage of PCs' tractability and still have an efficient implementation as a computation graph. Specifically, data softening provides a principled way to add uncertainty in datasets in closed form, which implicitly regularizes PC parameters. To learn parameters from a softened dataset, PCs only need linear time by virtue of their tractability. In entropy regularization, the exact entropy of the distribution encoded by a PC can be regularized directly, which is again infeasible for most other density estimation models. We show that both methods consistently improve the generalization performance of a wide variety of PCs. Moreover, when paired with a simple PC structure, we achieved state-of-the-art results on 10 out of 20 standard discrete density estimation benchmarks. Open-source code and experiments are available at https://github.com/UCLA-StarAI/Tractable-PC-Regularization.

## 1 Introduction

Probabilistic Circuits (PCs) [1, 2] are considered to be the lingua franca for Tractable Probabilistic Models (TPMs) as they offer a unified framework to abstract from a wide variety of TPM circuit representations, such as arithmetic circuits (ACs) [3], sum-product networks (SPNs) [4], and probabilistic sentential decision diagrams (PSDDs) [5]. PCs are a successful combination of classic probabilistic graphical models (PGMs) and neural networks (NNs). Moreover, by enforcing various structural properties, PCs permit efficient and exact computation of a large family of probabilistic inference queries [6, 7, 8]. The ability to answer these queries leads to successful applications in areas such as model compression [9] and model bias detection [10, 11]. At the same time, PCs are analogous to NNs since their evaluation is also carried out using computation graphs. By exploiting the parallel computation power of GPUs, dedicated implementations [2, 12] can train a complex PC with millions of parameters in minutes. These innovations have made PCs much more expressive and scalable to richer datasets that are beyond the reach of "older" TPMs [13].

35th Conference on Neural Information Processing Systems (NeurIPS 2021).

However, such advances make PCs more prone to overfitting. Although parameter regularization has been extensively studied in both the PGM and NN communities [14, 15], we find that existing regularization techniques for PGMs and NNs are either not suitable or not effective enough when applied to PCs. For example, parameter priors or Laplace smoothing typically used in PGMs, and often used in PC learning as well [16, 17, 18], incur unwanted bias when learning PC parameters – we will illustrate this point in Sec. 3. Classic NN methods such as L1 and L2 regularization are not always suitable since PCs often use either closed-form or EM-based parameter updates.

This paper designs parameter regularization methods that are directly tailored for PCs. We propose two regularization techniques, *data softening* and *entropy regularization*. Both formulate the regularization objective in terms of distributions, regardless of their representation and parameterization. Yet, both leverage the tractability and structural properties of PCs. Specifically, data softening injects noise into the dataset by turning hard evidence in the samples into soft evidence [19, 20]. While learning with such softened datasets is infeasible even for simple machine learning models, with their tractability, a class of PCs (i.e., *deterministic* PCs) can learn the maximum-likelihood estimation (MLE) parameters given a softened dataset in $\mathcal{O}(|p| \cdot |\mathcal{D}|)$ time, where $|p|$ is the size of the PC and $|\mathcal{D}|$ is the size of the (original) dataset. For PCs that are not deterministic, every parameter update step can be done in $\mathcal{O}(|p| \cdot |\mathcal{D}|)$ time, still allowing efficient parameter learning. Additionally, the entropy of the distribution encoded by a PC can be tractably regularized. Although the entropy regularization objective for PC is multi-modal and a global optimum cannot be found in general, we propose an algorithm that is guaranteed to converge monotonically towards a stationary point.

We show that both proposed approaches consistently improve the test set performance over standard density estimation benchmarks. Furthermore, we observe that when data softening and entropy regularization are properly combined, even better generalization performance can be achieved. Specifically, when paired with a simple PC structure, this combined regularization method achieves state-of-the-art results on 10 out of 20 standard discrete density estimation benchmarks.

**Notation**  We denote random variables by uppercase letters (e.g., $X$) and their assignments by lowercase letters (e.g., $x$). Analogously, we use bold uppercase letters (e.g., $\mathbf{X}$) and bold lowercase letters (e.g., $\boldsymbol{x}$) for sets of variables and their joint assignments, respectively.

## 2   Two Intuitive Ideas for Regularizing Distributions

A common way to prevent overfitting in machine learning models is to regularize the syntactic representation of the distribution. For example, L1 and L2 losses add mutually independent priors to all parameters of a model; other approaches such as Dropout [14], Bayesian Neural Networks (BNNs) [21], and Bayesian parameter smoothing [22] incorporate more complex and structured priors into the model [23]. In this section, we ask the question: how would we regularize an arbitrary distribution, regardless of the model at hand, and the way it is parameterized? Such global, model-agnostic regularizers appear to be under-explored. Next, we introduce two intuitive ideas for regularizing distributions, and study how they can be practically realized in the context of probabilistic circuits in the remainder of this paper.

**Data softening**  Data augmentation is a common technique to improve the generalization performance of machine learning models [24, 25]. A simple yet effective type of data augmentation is to inject noise into the samples, for example by randomly corrupting bits or pixels [26]. This can greatly improve generalization as it renders the model more robust to such noise. While current noise injection methods are implemented as a sequence of sampled transformations, we stress that some noise injection can be done in closed form: we will be considering all possible corruptions, each with their own probability, as a function of how similar they are to a training data point.

Consider boolean variables[1] as an example: after noise injection, a sample $X = 1$ is represented as a distribution over all possible assignments (i.e., $X = 1$ and $X = 0$), where the instance $X = 1$, which is "similar" to the original sample, gets a higher probability: $P(X = 1) = \beta$. Here $\beta \in (0.5, 1]$ is a hyperparameter that specifies the regularization strength — if $\beta = 1$, no regularization is added; if $\beta$ approaches 0.5, the regularized sample represents an (almost) uniform distribution. For a sample $\boldsymbol{x}$ with $K$ variables $\mathbf{X} := \{X_i\}_{i=1}^{K}$, where the $k$th variable takes value $x_k$, we can similarly 'soften' $\boldsymbol{x}$

---

[1]We postpone the discussion on regularizing samples with non-boolean variables in Appendix B.1.

by independently injecting noise into each variable, resulting in a *softened distribution* $P_{\boldsymbol{x},\beta}$:

$$\forall \boldsymbol{x}' \in \mathsf{val}(\mathbf{X}), \quad P_{\boldsymbol{x},\beta}(\mathbf{X} = \boldsymbol{x}') := \prod_{i=1}^{K} P_{\boldsymbol{x},\beta}(X_i = x_i') = \prod_{i=1}^{K} \left( \beta \cdot \mathbb{1}[x_i' = x_i] + (1-\beta) \cdot \mathbb{1}[x_i' \neq x_i] \right).$$

For a full dataset $\mathcal{D} := \{\boldsymbol{x}^{(i)}\}_{i=1}^{N}$, this softening of the data can also be represented through a new, *softened dataset* $\mathcal{D}_\beta$. Its empirical distribution is the average softened distribution of its data. It is a weighted dataset, where $\mathtt{weight}(\mathcal{D}_\beta, \boldsymbol{x})$ denotes the weight of sample $\boldsymbol{x}$ in $\mathcal{D}_\beta$:

$$\mathcal{D}_\beta := \{\boldsymbol{x} \,|\, \boldsymbol{x} \in \mathsf{val}(\mathbf{X})\} \quad \text{and} \quad \mathtt{weight}(\mathcal{D}_\beta, \boldsymbol{x}) = \frac{1}{N} \sum_{i=1}^{N} P_{\boldsymbol{x}^{(i)},\beta}(\mathbf{X} = \boldsymbol{x}). \tag{1}$$

This softened dataset ensures that each possible assignment has a small but non-zero weight in the training data. Consequently, any distribution learned on the softened data must assign a small probability everywhere as well. Of course, materializing this dataset, which contains all possible training example, is not practical. Regardless, we will think of data softening as implicitly operating on this softened dataset. We remark that data softening is related to soft evidence [27] and virtual evidence [28], which both define a framework to incorporate uncertain evidence into a distribution.

**Entropy regularization**  Shannon entropy is an effective indicator for overfitting. For a dataset $\mathcal{D}$ with $N$ distinct samples, a perfectly overfitting model that learns the exact empirical distribution has entropy $\log(N)$. A distribution that generalizes well should have a much larger entropy, since it assigns positive probability to exponentially more assignments near the training samples. Concretely, for the protein sequence density estimation task [29] that we will experiment with in Sec. 4.3, the perfectly overfitting empirical distribution has entropy 3, a severely overfitting learned model has entropy 92, yet a model that generalizes well has entropy 177. Therefore, directly controlling the entropy of the learned distribution will help mitigate overfitting. Given a model $P_{\boldsymbol{\theta}}$ parametrized by $\boldsymbol{\theta}$ and a dataset $\mathcal{D} := \{\boldsymbol{x}^{(i)}\}_{i=1}^{N}$, we define the following entropy regularization objective:

$$\mathrm{LL}_{\mathrm{ent}}(\boldsymbol{\theta}; \mathcal{D}, \tau) := \frac{1}{N} \sum_{i=1}^{N} \log P_{\boldsymbol{\theta}}(\boldsymbol{x}^{(i)}) + \tau \cdot \mathtt{ENT}(P_{\boldsymbol{\theta}}), \tag{2}$$

where $\mathtt{ENT}(P_{\boldsymbol{\theta}}) := -\sum_{\boldsymbol{x} \in \mathsf{val}(\mathbf{X})} P_{\boldsymbol{\theta}}(\boldsymbol{x}) \log P_{\boldsymbol{\theta}}(\boldsymbol{x})$ denotes the entropy of distribution $P_{\boldsymbol{\theta}}$, and $\tau$ is a hyperparameter that controls the regularization strength. Various forms of entropy regularization have been used in the training process of deep learning models. Different from Eq. (2), these methods regularize the entropy of a parametric [30, 31] or non-parametric [32] output space of the model.

Although both ideas for regularizing distributions are rather intuitive, it is surprisingly hard to implement them in practice since they are intractable even for the simplest machine learning models.

**Theorem 1.** *Computing the likelihood of a distribution represented as a exponentiated logistic regression (or equivalently, a single neuron) given softened data is #P-hard.*

**Theorem 2.** *Computing the Shannon entropy of a normalized logistic regression model is #P-hard.*

Proof of Thm. 1 and 2 are provided in Appendices A.3 and A.4. Although data softening and entropy regularization are infeasible for many models, we will show in the following sections that they are tractable to use when applied to Probabilistic Circuits (PCs) [1], a class of expressive TPMs.

## 3  Background and Motivation

Probabilistic Circuits (PCs) are a collective term for a wide variety of TPMs. They present a unified set of notations that provides succinct representations for TPMs such as Probabilistic Sentential Decision Diagrams (PSDDs) [5], Sum-Product Networks (SPNs) [4], and Arithmetic Circuits (ACs) [3]. We proceed by introducing the syntax and semantics of a PC.

**Definition 1** (Probabilistic Circuits)**.**  A PC $p$ that represents a probability distribution over variables $\mathbf{X}$ is defined by a parametrized directed acyclic graph (DAG) with a single root node, denoted $n_r$. The DAG comprises three kinds of units: *input*, *sum*, and *product*. Each leaf node $n$ in the DAG corresponds to an input unit; each inner node $n$ (i.e., sum and product units) receives inputs from its

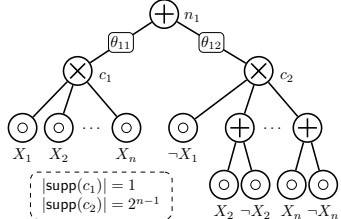 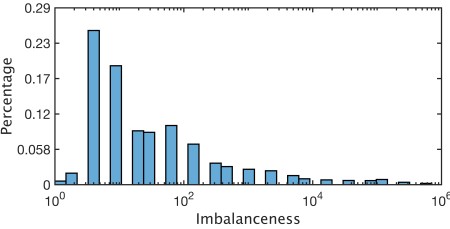

(a) A PC with imbalanced sum unit $n_1$.      (b) Imbalanceness of the PCs learned by Strudel.

Figure 1: A Problem of Laplace smoothing. (a) Laplace smoothing cannot properly regularize this PC as the sum unit $n_1$ is imbalanced, i.e., its two children have drastically different support sizes. (b) A large fraction of sum units learned by a PC structure learning algorithm [17] are imbalanced.

children, denoted $\mathsf{in}(n)$. Each unit $n$ encodes a probability distribution $p_n$, defined as follows:

$$
p_n(\boldsymbol{x}) := \begin{cases} f_n(\boldsymbol{x}) & \text{if } n \text{ is an input unit,} \\ \sum_{c \in \mathsf{in}(n)} \theta_{n,c} \cdot p_c(\boldsymbol{x}) & \text{if } n \text{ is a sum unit,} \\ \prod_{c \in \mathsf{in}(n)} p_c(\boldsymbol{x}) & \text{if } n \text{ is a product unit,} \end{cases}
$$

where $f_n$ is a univariate input distribution (e.g., boolean, categorical or Gaussian), and $\theta_{n,c}$ represents the parameter corresponds to edge $(n, c)$. Intuitively, a sum unit models a weighted mixture distribution over its children, and a product unit encodes a factored distribution over its children. We assume w.l.o.g. that all parameters are positive and the parameters associated with any sum unit $n$ sum up to 1 (i.e., $\sum_{c \in \mathsf{in}(n)} \theta_{n,c} = 1$). We further assume w.l.o.g. that a PC alternates between sum and product layers [33]. The size of a PC $p$, denoted $|p|$, is the number of edges in its DAG.

This paper focuses on two classes of PCs that support different types of queries: (i) PCs that allow linear-time computation of marginal (MAR) and maximum-a-posterior (MAP) inferences (e.g., PSDDs [5], selective SPNs [34]); (ii) PCs that only permit linear-time computation of MAR queries (e.g., SPNs [4]). The borders between these two types of PCs are defined by their *structural properties*, i.e., constraints imposed on a PC. First, in order to compute MAR queries in linear time, both classes of PCs should be decomposable (Def. 2) and smooth (Def. 3) [1]. These are properties of the (variable) scope $\phi(n)$ of PC units $n$, that is, the collection of variables defined by all its descendent input nodes.

**Definition 2** (Decomposability). A PC is decomposable if for every product unit $n$, its children have disjoint scopes: $\forall c_1, c_2 \in \mathsf{in}(n) \ (c_1 \neq c_2), \phi(c_1) \cap \phi(c_2) = \varnothing$.

**Definition 3** (Smoothness). A PC is smooth if for every sum unit $n$, its children have the same scope: $\forall c_1, c_2 \in \mathsf{in}(n), \phi(c_1) = \phi(c_2)$.

Next, *determinism* is required to guarantee efficient computation of MAP inference [35].

**Definition 4** (Determinism). Define the support $\mathsf{supp}(n)$ of a PC unit $n$ as the set of complete variable assignments $\boldsymbol{x} \in \mathsf{val}(\mathbf{X})$ for which $p_n(\boldsymbol{x})$ has non-zero probability (density): $\mathsf{supp}(n) = \{\boldsymbol{x} \mid \boldsymbol{x} \in \mathsf{val}(\mathbf{X}), p_n(\boldsymbol{x}) > 0\}$. A PC is deterministic if for every sum unit $n$, its children have disjoint support: $\forall c_1, c_2 \in \mathsf{in}(n) \ (c_1 \neq c_2), \mathsf{supp}(c_1) \cap \mathsf{supp}(c_2) = \varnothing$.

Since the only difference in the structural properties of both PCs classes is determinism, we denote members in the first PC class as deterministic PCs, and members in the second PC class as non-deterministic PCs. Interestingly, both PC classes not only differ in their tractability, which is characterized by the set of queries that can be computed within $\mathtt{poly}(|p|)$ time [6], they also exhibit drastically different expressive efficiency. Specifically, abundant empirical [17, 13] and theoretical [36] evidences suggest that non-deterministic PCs are more expressive than their deterministic counterparts. Due to their differences in terms of tractability and expressive efficiency, this paper studies parameter regularization on deterministic and non-deterministic PCs separately.

**Motivation** Laplace smoothing is widely adopted as a PC regularizer [16, 17]. Since it is also the default regularizer for classical probabilistic models such as Bayesian Networks (BNs) [37] and Hierarchical Bayesian Models (HBMs) [38], this naturally raises the following question: *are there differences between a good regularizer for classical probabilistic models such as BNs and HBMs and effective regularizers for PCs?* The question can be answered affirmatively — while Laplace

| **Algorithm 1** Forward pass | **Algorithm 2** Backward pass |
|---|---|
| 1: **Input:** A deterministic PC $p$; sample $\boldsymbol{x}$ 
 2: **Output:** $\texttt{value}[n] := (\boldsymbol{x} \in \mathsf{supp}(n))$ for each unit $n$ 
 3: **foreach** $n$ traversed in postorder **do** 
 4:    **if** $n$ **isa** input unit **then** $\texttt{value}[n] \leftarrow f_n(\boldsymbol{x})$ 
 5:    **elif** $n$ **isa** product unit **then** 
 6:      $\lfloor \texttt{value}[n] \leftarrow \prod_{c \in \mathsf{in}(n)} \texttt{value}[c]$ 
 7:    **else** //$n$ is a sum unit 
 8:      $\lfloor \texttt{value}[n] \leftarrow \sum_{c \in \mathsf{in}(n)} \texttt{value}[c]$ | 1: **Input:** A deterministic PC $p$; $\forall n, \texttt{value}[n]$ 
 2: **Output:** $\texttt{flow}[n,c] := (\boldsymbol{x} \in (\gamma_n \cap \gamma_c))$ for each 
     pair $(n,c)$, where $n$ is a sum unit and $c \in \mathsf{in}(n)$ 
 3: $\forall n, \texttt{context}[n] \leftarrow 0; \texttt{context}[n_r] \leftarrow \texttt{value}[n_r]$ 
 4: **foreach** sum unit $n$ traversed in preorder **do** 
 5:    **foreach** $m \in \mathsf{pa}(n)$ **do**    (denote $g \leftarrow \mathsf{pa}(m)$) 
 6:      $\texttt{f} \leftarrow \frac{\texttt{value}[m]}{\texttt{value}[g]} \cdot \texttt{context}[g]$ 
 7:      $\texttt{context}[n] \mathrel{+}= \texttt{f}; \quad \texttt{flow}[g,m] = \texttt{f}$ |

smoothing provides good priors to BNs and HBMs, its uniform prior could add unwanted bias to PCs. Specifically, for every sum unit $n$, Laplace smoothing assigns the same prior to all its child parameters (i.e., $\{\theta_{n,c} \mid c \in \mathsf{in}(n)\}$), while in many practical PCs, these parameters should be given drastically different priors. For example, consider the PC shown in Fig. 1(a). Since $c_2$ has an exponentially larger support than $c_1$, it should be assumed as prior that $\theta_{12}$ will be much larger than $\theta_{11}$.

We highlight the significance of the above issue by examining the fraction of sum units with imbalanced child support sizes in PCs learned by Strudel, a state-of-the-art structure learning algorithm for deterministic PCs [5]. We examine 20 PCs learned from the 20 density estimation benchmarks [39], respectively. All sum units with $\geq 3$ children and with a support size $\geq 128$ are recorded. We measure "imbalanceness" of a sum unit $n$ by the fraction of the maximum and minimum support size of its children (i.e., $\frac{\max_{c_1 \in \mathsf{in}(n)} |\mathsf{supp}(c_1)|}{\min_{c_2 \in \mathsf{in}(n)} |\mathsf{supp}(c_2)|}$). As demonstrated in Fig. 1(b), more than $20\%$ of the sum units have imbalanceness $\geq 10^2$, which suggests that the inability of Laplace smoothing to properly regularize PCs with imbalanced sum units could lead to severe performance degradation in practice.

# 4 How Is This Tractable And Practical?

In this section, we first provide additional background about the parameter learning algorithms for deterministic and non-deterministic PCs (Sec. 4.1). We then demonstrate how the two intuitive ideas for regularizing distributions (Sec. 2), i.e., data softening and entropy regularization, can be efficiently implemented for deterministic (Sec. 4.2) and non-deterministic (Sec. 4.3) PCs.

## 4.1 Learning the Parameters of PCs

**Deterministic PCs** Given a deterministic PC $p$ defined on variables $\mathbf{X}$ and a dataset $\mathcal{D} = \{\boldsymbol{x}^{(i)}\}_{i=1}^{N}$, the maximum likelihood estimation (MLE) parameters $\boldsymbol{\theta}_{\mathcal{D}}^{*} := \mathrm{argmax}_{\boldsymbol{\theta}} \sum_{i=1}^{N} \log p(\boldsymbol{x}^{(i)}; \boldsymbol{\theta})$ can be learned in closed-form. To formalize the MLE solution, we need a few extra definitions.

**Definition 5** (Context). The context $\gamma_n$ of every unit $n$ in a PC $p$ is defined in a top-down manner: for the base case, context of the root node $n_r$ is defined as its support: $\gamma_{n_r} := \mathsf{supp}(n_r)$. For every other node $n$, its context is the intersection of its support and the union of its parents' ($\mathsf{pa}(n)$) contexts:

$$\gamma_n := \bigcup_{m \in \mathsf{pa}(n)} \gamma_m \cap \mathsf{supp}(n).$$

Intuitively, if an assignment $\boldsymbol{x}$ is in the context of unit $n$, then there exists a path on the PC's DAG from $n$ to the root unit $n_r$ such that for any unit $m$ in the path, we have $\boldsymbol{x} \in \mathsf{supp}(m)$. Circuit flow extends the notation of context to indicate whether a sample $\boldsymbol{x}$ is in the context of an edge $(n,c)$.

**Definition 6** (Flows). The flow $\mathrm{F}_{n,c}(\boldsymbol{x})$ of any edge $(n,c)$ in a PC given variable assignments $\boldsymbol{x} \in \mathsf{val}(\mathbf{X})$ is defined as $\mathbb{1}[\boldsymbol{x} \in \gamma_n \cap \gamma_c]$, where $\mathbb{1}[\cdot]$ is the indicator function. The flow $\mathrm{F}_{n,c}(\mathcal{D})$ w.r.t. dataset $\mathcal{D} = \{\boldsymbol{x}^{(i)}\}_{i=1}^{N}$ is the sum of the flows of all its samples: $\mathrm{F}_{n,c}(\mathcal{D}) := \sum_{i=1}^{N} \mathrm{F}_{n,c}(\boldsymbol{x}^{(i)})$.

The flow $\mathrm{F}_{n,c}(\boldsymbol{x})$ for *all* edges $(n,c)$ in a PC $p$ w.r.t. sample $\boldsymbol{x}$ can be computed through a forward and backward path that both take $\mathcal{O}(|p|)$ time. The forward path, as shown in Alg. 1, starts from the leaf units and traverses the PC in postorder to compute $\forall n, \texttt{value}[n] := \mathbb{1}[\boldsymbol{x} \in \mathsf{supp}(n)]$; afterwards, the backward path illustrated in Alg. 2 begins at the root unit $n_r$ and traverses the PC in preorder to

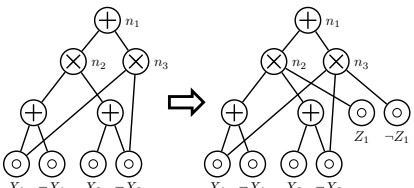
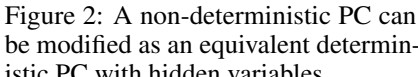

Figure 2: A non-deterministic PC can be modified as an equivalent deterministic PC with hidden variables.

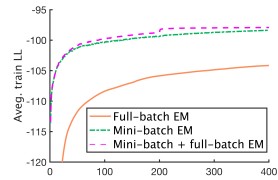

Figure 3: Average train LL on MNIST using different EM updates.

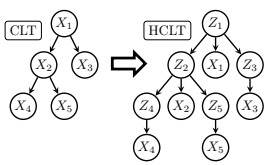

Figure 4: HCLT is constructed by adding hidden variables in a CLT [43].

compute $\forall n$, $\mathtt{context}[n] := \mathbb{1}[\boldsymbol{x} \in \gamma_n]$ as well as $\forall(n,c)$, $\mathtt{flow}[n,c] := \mathrm{F}_{n,c}(\boldsymbol{x})$. By Def. 6, the time complexity for computing $\mathrm{F}_{n,c}(\mathcal{D})$ with respect to all edges $(n,c)$ in $p$ is $\mathcal{O}(|p| \cdot |\mathcal{D}|)$, where $|\mathcal{D}|$ is the size of dataset $\mathcal{D}$. The correctness of Alg. 1 and 2 are justified in Appendix A.6.

The MLE parameters $\boldsymbol{\theta}_{\mathcal{D}}^*$ given dataset $\mathcal{D}$ can be computed using the flows [5]:

$$\forall(n,c), \quad \theta_{n,c}^* = \mathrm{F}_{n,c}(\mathcal{D}) / \sum\nolimits_{c \in \mathsf{in}(n)} \mathrm{F}_{n,c}(\mathcal{D}). \tag{3}$$

Define hyperparameter $\alpha$ ($\alpha \geq 0$), for every sum unit $n$, Laplace smoothing regularizes its child parameters (i.e., $\{\theta_{n,c} \mid c \in \mathsf{in}(n)\}$) by adding a *pseudocount* $\alpha/|\mathsf{in}(n)|$ to every child branch of $n$, which is equivalent to adding $\alpha/|\mathsf{in}(n)|$ to the numerator of Eq. (3) and $\alpha$ to its denominator.

**Non-deterministic PCs** As justified by Peharz et al. [40], every non-deterministic PC can be augmented as a deterministic PC with additional hidden variables. For example, in Fig. 2, the left PC is not deterministic since the support of both children of $n_1$ (i.e., $n_2$ and $n_3$) contains $x_1 \bar{x}_2$. The right PC augments the left one by adding input units correspond to hidden variable $Z_1$, which retains determinism by "dividing" the overlapping support $x_1 \bar{x}_2$ into $x_1 \bar{x}_2 z_1 \in \mathsf{supp}(n_2)$ and $x_1 \bar{x}_2 \bar{z}_1 \in \mathsf{supp}(n_3)$. Under this interpretation, parameter learning of non-deterministic PCs is equivalent to learning the parameters of deterministic PCs given incomplete data (we never observe the hidden variables), which can be solved by Expectation-Maximization (EM) [41, 42]. In fact, EM is the default parameter learning algorithm for non-deterministic PCs [13, 10].

Under the latent variable model view of a non-deterministic PC, its EM updates can be computed using *expected flows* [10]. Specifically, given observed variables $\mathbf{X}$ and (implicit) hidden variables $\mathbf{Z}$, the expected flow of edge $(n,c)$ given dataset $\mathcal{D}$ is defined as

$$\mathrm{EF}_{n,c}(\mathcal{D}; \boldsymbol{\theta}) := \mathbb{E}_{\boldsymbol{x} \sim \mathcal{D}, \boldsymbol{z} \sim p_c(\cdot \mid \boldsymbol{x}; \boldsymbol{\theta})}[\mathrm{F}_{n,c}(\boldsymbol{x}, \boldsymbol{z})],$$

where $\boldsymbol{\theta}$ is the set of parameters, and $p_c(\cdot \mid \boldsymbol{x}; \boldsymbol{\theta})$ is the conditional probability over hidden variables $\mathbf{Z}$ given $\boldsymbol{x}$ specified by the PC rooted at unit $c$. Similar to flows, the expected flows can be computed via a forward and backward pass of the PC (Alg. 5 and 6 in the Appendix). As shown by Choi et al. [10], for a non-deterministic PC, its parameters for the next EM iteration are given by

$$\theta_{n,c}^{(new)} = \mathrm{EF}_{n,c}(\mathcal{D}; \boldsymbol{\theta}) / \sum_{c \in \mathsf{in}(n)} \mathrm{EF}_{n,c}(\mathcal{D}; \boldsymbol{\theta}). \tag{4}$$

This paper uses a hybrid EM algorithm, which uses mini-batch EM updates to initiate the training process, and switch to full-batch EM updates afterwards. Specifically, in mini-batch EM, $\boldsymbol{\theta}^{(new)}$ are computed using mini-batches of samples, and the parameters are updated towards the taget with a step size $\eta$: $\boldsymbol{\theta}^{(k+1)} \leftarrow (1-\eta)\boldsymbol{\theta}^{(k)} + \eta\boldsymbol{\theta}^{(new)}$; when using full-batch EM, we iteratively compute the updated parameters $\boldsymbol{\theta}^{(new)}$ using the whole dataset. Fig. 3 demonstrates that this hybrid approach offers faster convergence speed compared to using full-batch or mini-batch EM only.

### 4.2 Regularizing Deterministic PCs

We demonstrate how the intuitive ideas for regularizing distributions presented in Sec. 2 (i.e., data softening and entropy regularization) can be efficiently applied to deterministic PCs.

**Data softening** As hinted by Eq. (1), we need exponentially many samples to represent a softened dataset, which makes parameter learning intractable even for the simple logistic regression model (Thm. 1), let alone more complex probabilistic models such as VAEs [44] and GANs [45]. Despite

---
**Algorithm 3** PC Entropy regularization
---
1: **Input:** A deterministic PC $p$; flow $\mathrm{F}_{n,c}(\mathcal{D})$ for every edge $(n,c)$ in $p$; hyperparameter $\tau$.
2: **Output:** A set of log-parameters, $\{\varphi_{n,c} : (n,c) \in p\}$, which are the solution of Eq. (2).
3: $\forall n$, $\texttt{node\_prob}[n] \leftarrow 0$; $\quad \texttt{node\_prob}[n_r] \leftarrow 1 \quad //n_r$ is the root node of $p$
4: **while** not converge **do**
5: $\quad \forall n$, $\texttt{entropy}[n] \leftarrow$ The entropy of the sub-PC rooted at $n$ (see Alg. 4 in Appendix A.2)
6: $\quad$ **foreach** sum unit $n$ traversed in preorder (parent before children) **do**
7: $\qquad d_i \leftarrow \mathrm{F}_{n,c_i}(\mathcal{D})/|\mathcal{D}|; \quad b = \tau \cdot \texttt{node\_prob}[n] \quad //\{c_i\}_{i=1}^{\mathsf{in}(n)}$ is the set of children of $n$
8: $\qquad$ Solve for $\{\varphi_{n,c_i}\}_{i=1}^{|\mathsf{in}(n)|}$ in the following set of equations ($y$ is a variable):

$$\begin{cases} d_i e^{-\varphi_{n,c_i}} - b \cdot \varphi_{n,c_i} + b \cdot \texttt{entropy}[c_i] = y & (\forall i \in \{1, \dots, |\mathsf{in}(n)|\}) \\ \sum_{i=1}^{|\mathsf{in}(n)|} e^{\varphi_{n,c_i}} = 1 \end{cases} \tag{5}$$

9: $\qquad$ **for** each $c \in \mathsf{in}(n)$ and each $m \in \mathsf{in}(c)$ **do** //Update $\texttt{node\_prob}$ of grandchildren
10: $\qquad\quad \lfloor \texttt{node\_prob}[m] \leftarrow \texttt{node\_prob}[m] + e^{\varphi_{n,c}} \cdot \texttt{node\_prob}[n]$
---

this negative result, the MLE parameters of a PC $p$ w.r.t. $\mathcal{D}_\beta$ can be computed in time $\mathcal{O}(|p|\cdot|\mathcal{D}|)$, which is linear w.r.t. the model size as well as the size of the *original* dataset.

**Theorem 3.** *Let $f_n(\boldsymbol{x}) = \beta \cdot \mathbb{1}[\boldsymbol{x} \in \mathsf{supp}(n)] + (1-\beta) \cdot \mathbb{1}[\boldsymbol{x} \notin \mathsf{supp}(n)]$ in Alg. 1. Given a deterministic PC $p$, a boolean dataset $\mathcal{D}$, and hyperparameter $\beta \in (0.5, 1]$, the set of all flows $\{\mathrm{F}_{n,c}(\mathcal{D}_\beta) \mid \forall \text{ edge } (n,c)\}$ w.r.t. the softened dataset $\mathcal{D}_\beta$ can be computed by Alg. 1 and 2 within $\mathcal{O}(|p|\cdot|\mathcal{D}|)$ time.*

Proof of this theorem is provided in Appendix A.1. Since the MLE parameters (Eq. (3)) w.r.t. $\mathcal{D}_\beta$ can be computed in $\mathcal{O}(|p|)$ time using the flows, the overall time complexity to compute the MLE parameters is again $\mathcal{O}(|p|\cdot|\mathcal{D}|)$.

**Entropy regularization** The hope for tractable PC entropy regularization comes from the fact that the entropy of a deterministic PC $p$ can be exactly computed in $\mathcal{O}(|p|)$ time [6, 46]. However, it is still unclear whether the entropy regularization objective $\mathrm{LL}_{\mathrm{ent}}(\boldsymbol{\theta}; \mathcal{D}, \tau)$ (Eq. (2)) can be tractably maximized. We answer this question with a mixture of positive and negative results: while the objective is multi-modal and the global optimal is hard to find, we propose an efficient algorithm that (i) guarantees convergence to a stationary point, and (ii) achieves high convergence rate in practice. We start with the negative result.

**Proposition 1.** *There exists a deterministic PC $p$, a hyperparameter $\tau$, and a dataset $\mathcal{D}$ such that $\mathrm{LL}_{\mathrm{ent}}(\boldsymbol{\theta}; \mathcal{D}, \tau)$ (Eq. (2)) is non-concave and has multiple local maximas.*

Proof is given in Appendix A.7. Although global optimal solutions are generally infeasible, we propose an efficient algorithm that guarantees to find a stationary point of $\mathrm{LL}_{\mathrm{ent}}(\boldsymbol{\theta}; \mathcal{D}, \tau)$. Specifically, Alg. 3 takes as input a deterministic PC $p$ and all its edge flows w.r.t. $\mathcal{D}$, and returns a set of learned log-parameters that correspond to a stationary point of the objective.[2] In its main loop (lines 4-10), the algorithm alternates between two procedures: (i) compute the entropy of the distribution encoded by every node w.r.t. the current parameters (line 5),[3] and (ii) update PC parameters with regard to the computed entropies (lines 6-10). Specifically, in the parameter update phase (i.e., the second phase), the algorithm traverses every sum unit $n$ in preorder and updates its child parameters by maximizing the entropy regularization objective ($\mathrm{LL}_{\mathrm{ent}}(\boldsymbol{\theta}; \mathcal{D}, \tau)$) with all other parameters fixed. This is done by solving the set of equations in Eq. (5) using Newton's method (lines 7-8).[4] In addition to the child nodes' entropy computed in the first phase, Eq. (5) uses the top-down probability of every unit $n$ (i.e., $\texttt{node\_prob}[n]$), which is progressively updated in lines 9-10.

**Theorem 4.** *Alg. 3 converges monotonically to a stationary point of $\mathrm{LL}_{\mathrm{ent}}(\boldsymbol{\theta}; \mathcal{D}, \tau)$ (Eq. (2)).*

*Proof.* The high-level idea of the proof is to show that the parameter update phase (lines 6-10) optimizes a concave surrogate objective of $\mathrm{LL}_{\mathrm{ent}}(\boldsymbol{\theta}; \mathcal{D}, \tau)$, which is determined by the entropies computed

---
[2] We compute parameters in the logarithm space for numerical stability.
[3] This can be done by Alg. 4 shown in Appendix A.2. Lem. 1 proves that Alg. 4 takes $\mathcal{O}(|p|)$ time.
[4] Details for solving Eq. (5) is given in Appendix B.2.

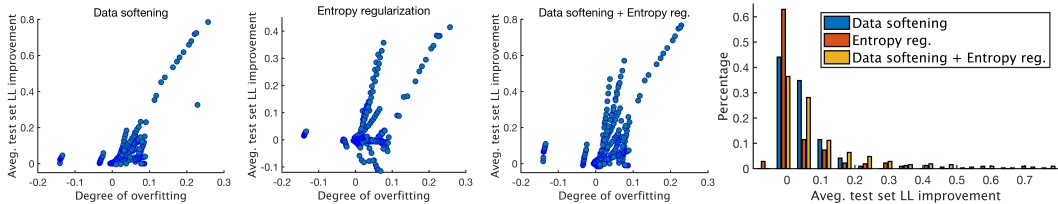

Figure 5: Both data softening and entropy regularization effectively improve the test set log-likelihood (LL) across various datasets [39] and PC structures [17]. LL improvement (higher is better) represents the gain of test set LL compared to Laplace smoothing. The test-set LLs are reported in Table 3.

in line 5. Specifically, we show that whenever the surrogate objective is improved, $\text{LL}_{\text{ent}}(\boldsymbol{\theta}; \mathcal{D}, \tau)$ is also improved. Since the surrogate objective is concave, it can be easily optimized. Therefore, Alg. 3 converges to a stationary point of $\text{LL}_{\text{ent}}(\boldsymbol{\theta}; \mathcal{D}, \tau)$. The detailed proof is in Appendix A.5. □

Alg. 3 can be regarded as a EM-like algorithm, where the E-step is the entropy computation phase (line 5) and the M-step is the parameter update phase (lines 6-10). Specifically, the E-step constructs a concave surrogate of the true objective ($\text{LL}_{\text{ent}}(\boldsymbol{\theta}; \mathcal{D}, \tau)$), and the M-step updates all parameters by maximizing the concave surrogate function. Although Thm. 4 provides no convergence rate analysis, the outer loop typically takes 3-5 iterations to converge in practice. Furthermore, Eq. (5) can be solved with high precision in a few ($< 10$) iterations. Therefore, compared to the computation of all flows w.r.t. $\mathcal{D}$, which takes $\mathcal{O}(|p| \cdot |\mathcal{D}|)$ time, Alg. 3 takes a negligible $\mathcal{O}(|p|)$ time.

In response to the motivation in Sec. 3, we show that both proposed methods can overcome the imbalanced regularization problem of Laplace smoothing. Again consider the example PC in Fig. 1(a), we conceptually demonstrate that both data softening and entropy regularization will not over-regularization $\theta_{11}$ compared to $\theta_{12}$. First, data softening essentially add no prior to the parameters, and only soften the evidences in the dataset. Therefore, it will not over-regularize children with small support sizes. Second, entropy regularization will add a much higher prior to $\theta_{12}$. Suppose $n = 10$, consider maximizing Eq. (2) with an empty dataset (i.e., we maximize $\text{ENT}(p_{n_1})$ directly), the optimal parameters would be $\theta_{11} \approx 0.002$ and $\theta_{12} \approx 0.998$. Therefore, entropy regularization will tend to add a higher prior to children with large support sizes. More fundamentally, the reason why both proposed approaches do not add biased priors to PCs is that they are designed to be model-agnostic, i.e., their definitions as shown in Sec. 2 are independent with the model they apply to.

**Empirical evaluation** We empirically evaluate both proposed regularization methods on the twenty density estimation datasets [39]. Since we are only concerned with parameter learning, we adopt PC structures (defined by its DAG) learned by Strudel [17]. 16 PCs with different sizes were selected for each of the 20 datasets. For all experiments, we performed a hyperparameter search for all three regularization approaches (Laplace smoothing, data softening, and entropy regularization)[5] using the validation set and report results on the test set. Please refer to Appendix B.3 for more details.

Results are summarized in Fig. 5. First look at the scatter plots on the left. The x-axis represents the degree of overfitting, which is computed as follows: denote $\text{LL}_{train}$ and $\text{LL}_{val}$ as the average train and validation log-likelihood under the MLE estimation with Laplace smoothing ($\alpha = 1.0$), the degree of overfitting is defined as $(\text{LL}_{val} - \text{LL}_{train})/\text{LL}_{val}$, which roughly captures how much the dataset/model pair suffers from overfitting. The y-axis represents the improvement on the average test set log-likelihood compared to Laplace smoothing. As demonstrated by the scatter plots, despite a few outliers, both proposed regularization methods steadily improve the test set LL over various datasets and PC structures, and the LL improvements are positively correlated with the degree of overfitting. Furthermore, as shown by the last scatter plot and the histogram plot, when combining data softening and entropy regularization, the LL improvement becomes much higher compared to using the two regularizers individually.

### 4.3 Regularizing Non-Deterministic PCs

By viewing every non-deterministic PC as a deterministic PC with additional hidden variables (Sec. 4.1), the regularization techniques developed in Sec. 4.2 can be directly adapted. Specifically,

---

[5]Specifically, $\alpha \in \{0.1, 0.4, 1.0, 2.0, 4.0, 10.0\}$, $\beta \in \{0.9996, 0.999, 0.996\}$, $\tau \in \{0.001, 0.01, 0.1\}$.

Table 1: Test set log-likelihood in 20 density estimation benchmarks. We compare our method (HCLT) with the best performance (Best PSDD) over 2 deterministic PC learner: Strudel [17] and LearnPSDD [16] as well as the best performance (Best SPN) over 4 SPN learning algorithms: EinSumNet [13], LearnSPN [18], ID-SPN [47], and RAT-SPN [48]. With the help of data softening and entropy regularization ($\alpha = 0.1$, $\beta = 0.998$, and $\tau = 0.001$), HCLT achieved the best performance over 10 out of 20 datasets. All experiments for HCLT were repeated 5 times, and the average and standard deviation are reported.

| Dataset | HCLT | Best PSDD | Best SPN | Dataset | HCLT | Best PSDD | Best SPN |
|---------|------|-----------|----------|---------|------|-----------|----------|
| accidents | **-26.74**±0.03 | -28.29 | -26.98 | jester | **-52.46**±0.01 | -54.63 | -52.56 |
| ad | **-16.07**±0.06 | -16.52 | -19.00 | kdd | -2.18±0.00 | -2.17 | **-2.12** |
| baudio | **-39.77**±0.01 | -41.51 | -39.79 | kosarek | -10.66±0.01 | -10.98 | **-10.60** |
| bbc | -251.04±1.19 | -258.96 | **-248.33** | msnbc | -6.05±0.01 | -6.04 | **-6.03** |
| bnetflix | **-56.27**±0.01 | -58.53 | -56.36 | msweb | -9.98±0.05 | -9.93 | **-9.73** |
| book | **-33.83**±0.01 | -35.77 | -34.14 | nltcs | **-5.99**±0.01 | -6.03 | -6.01 |
| c20ng | -153.40±3.83 | -160.43 | **-151.47** | plants | -14.26±0.16 | -13.49 | **-12.54** |
| cr52 | -86.26±3.67 | -92.38 | **-83.35** | pumbs* | -23.64±0.25 | -25.28 | **-22.40** |
| cwebkb | -152.77±1.07 | -160.5 | **-151.84** | tmovie | **-50.81**±0.12 | -55.41 | -51.51 |
| dna | **-79.05**±0.17 | -82.03 | -81.21 | tretail | **-10.84**±0.01 | -10.90 | -10.85 |

data softening can be regarded as injecting noise in both observed and hidden variables. Since the dataset provides no information about the hidden variables anyway, data softening essentially still "perturbs" the observed variables only. On the other hand, entropy regularization will have different behaviors when applied to non-deterministic PCs. Specifically, since it is coNP-hard to compute the entropy of a non-deterministic PC [6], it is infeasible to optimize the entropy regularization objective $\text{LL}_{\text{ent}}(\boldsymbol{\theta}; \mathcal{D}, \tau)$ (Eq. (2)). However, we can still regularize the entropy of the distribution encoded by a non-deterministic PC over both of its observed and hidden variables, since explicitly representing the hidden variables renders the PC deterministic (Sec. 4.1).

On the implementation side, data softening is performed by modifying the forward pass of the algorithm used to compute expected flows (i.e., Alg. 5 and 6 in the Appendix). Entropy regularization is again performed by Alg. 3 at the M-step of each min-batch/full-batch EM update, except that the input flows (i.e., F) are replaced by the corresponding expected flows (i.e., EF).

**Empirical evaluation** We use a simple yet effective PC structure, hidden Chow-Liu Tree (HCLT), as demonstrated in Fig. 4. Specifically, on the left is a Bayesian network representation of a Chow-Liu Tree (CLT) [43] over 5 variables. For any CLT over variables $\{X_i\}_{i=1}^{k}$, we can modify it as a HCLT through the following steps. First, we introduce a set of $k$ latent variables $\{Z_i\}_{i=1}^{k}$. Next, we replace all observed variables in the CLT with its corresponding latent variable (i.e., $\forall i, X_i$ is replaced by $Z_i$). Finally, we add an edge from every latent variable to its corresponding observed variable (i.e., $\forall i$, add an edge $Z_i \rightarrow X_i$). The HCLT structure is then compiled into a PC that encodes the same probability distribution. We used

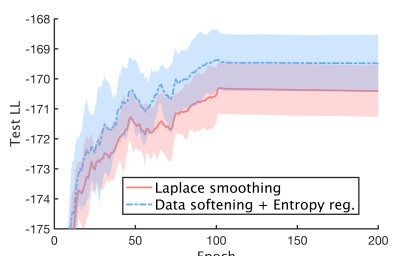

Figure 6: Average (±std) test LL over 5 trials on the protein dataset.

the hybrid mini-batch + full-batch EM as described in Sec. 4.1. For all experiments, we trained the PCs with 100 mini-batch EM epochs and 100 full-batch EM epochs. Please refer to Appendix B.4 for hyperparameters related to the HCLT structure and the parameter learning process. Similar to Sec. 4.2, we perform hyperparameter search for all methods using the validation set, and report results on the test set.

We first examine the performance on a protein sequence dataset [29] that suffers from severe overfitting. Specifically, the training LL is typically above $-100$ while the validation and test set LL are around $-170$. Fig. 6 shows the test LL for Laplace smoothing and the hybrid regularization approach as training progresses. With the help of data softening and entropy regularization, we were able to obtain consistently higher test set LL. Next, we compare our HCLT model (with regularization) with the state-of-the-art PSDD (Strudel [17] and LearnPSDD [16]) and SPN (EinSumNet [13], LearnSPN [18], ID-SPN [47], and RAT-SPN [48]) learning algorithms. Results are summarized in Table 1. With proper regularization, HCLT out-performed all baselines in 10 out of 20 datasets. Comparing with individual baselines, HCLT out-performs both PSDD learners on all datasets; HCLT

achieved higher log-likelihood on 18, 19, 10, and 17 datasets compared to EinSumNet, LearnSPN, ID-SPN, and RAT-SPN, respectively.

# 5  Conclusions

This paper proposes two model-agnostic distribution regularization techniques: data softening and entropy regularization. While both methods are infeasible for many machine learning models, we theoretically show that they can be efficiently implemented when applied to probabilistic circuits. On the empirical side, we show that both proposed regularizers consistently improve the generalization performance over a wide variety of PC structures and datasets.

**Acknowledgement**   This work is partially supported by NSF grants #IIS-1943641, #IIS-1956441, #CCF-1837129, DARPA grant #N66001-17-2-4032, a Sloan Fellowship, Intel, and Facebook.

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
