# Supplementary Material

## A  Proofs

This section provides the full proof of the theorems stated in the main paper.

### A.1  Proof of Theorem 3

**High-level idea**  The high-level idea of this proof is by separately showing the correctness of the forward pass (Alg. 1) and the backward pass (Alg. 2). Specifically, for a "softened" sample $\boldsymbol{x}$, we aim to show that (i) in the forward pass, the value of $\boldsymbol{x}$ w.r.t. any PC unit $n$ corresponds to the likelihood of $\boldsymbol{x}$ (note that since $\boldsymbol{x}$ can be represented as a weighted sum of exponentially many "hard" samples, the target likelihood is also the weighted sum of the respective likelihoods), and (ii) in the backward pass, the flow of $\boldsymbol{x}$ w.r.t. any PC unit corresponds to the weighted sum of the flows of the "hard" samples "contained" in $\boldsymbol{x}$. Both claims are proved by induction: for the forward pass, we first show that the base cases (leaf nodes) satisfy the claim, then by assuming all children of a PC unit satisfy the claim, we prove the inductive case of sum and product units; for the backward pass, induction is also applied in the preorder (parents before children).

As stated in the theorem, assume that we are given a deterministic PC $p$, a boolean dataset $\mathcal{D}$ containing $N$ samples $\{\boldsymbol{x}^{(i)}\}_{i=1}^{N}$, and hyperparameter $\beta \in (0.5, 1]$. Define $K$ as the number of variables in $\mathbf{X}$, i.e., $\mathbf{X} = \{X_k\}_{k=1}^{K}$.

**Correctness of the forward pass**  We show that the value of each node $n$ w.r.t. sample $\boldsymbol{x}^{(i)}$ (by slightly abusing notation, denoted as $\texttt{value}_i[n]$) computed by Alg. 1 (with the specific choice of $f_n(\boldsymbol{x}) = \beta \cdot \mathbb{1}[\boldsymbol{x} \in \texttt{supp}(n)] + (1 - \beta) \cdot \mathbb{1}[\boldsymbol{x} \notin \texttt{supp}(n)]$) is defined as

$$\texttt{value}_i[n] = \sum_{\boldsymbol{x} \in \texttt{val}(\mathbf{X})} \prod_{k=1}^{K} \left( \beta \cdot \mathbb{1}[x_k^{(i)} = x_k] + (1-\beta) \cdot \mathbb{1}[x_k^{(i)} \neq x_k] \right) \cdot \mathbb{1}[\boldsymbol{x} \in \texttt{supp}(n)], \quad (6)$$

where $x_k$ denotes the $k$th feature of $\boldsymbol{x}$.

• Base case: input units. Suppose node $n$ is a literal w.r.t. variable $X_k$. That is, $\boldsymbol{x} \in \texttt{supp}(n)$ iff $x_k = \texttt{Lit}(n)$, where $\texttt{Lit}(n)$ is either $\texttt{true}$ or $\texttt{false}$ defined by the PC. Denote $\neg\texttt{Lit}(n)$ as the negation of $\texttt{Lit}(n)$. $\forall i \in \{1, \ldots, N\}$ we have

$$\texttt{value}_i[n] = \beta \cdot \mathbb{1}[\boldsymbol{x}^{(i)} \in \texttt{supp}(n)] + (1 - \beta) \cdot \mathbb{1}[\boldsymbol{x}^{(i)} \notin \texttt{supp}(n)]$$

$$= \beta \cdot \mathbb{1}[x_k^{(i)} = \texttt{Lit}(n)] + (1 - \beta) \cdot \mathbb{1}[x_k^{(i)} = \neg\texttt{Lit}(n)]$$

$$\overset{(a)}{=} \sum_{\boldsymbol{x} \in \{\boldsymbol{x} : \boldsymbol{x} \in \texttt{val}(\mathbf{X}) \wedge x_k = \texttt{Lit}(n)\}} \prod_{l=1, l \neq k}^{K} \left( \beta \cdot \mathbb{1}[x_l^{(i)} = x_l] + (1-\beta) \cdot \mathbb{1}[x_l^{(i)} \neq x_l] \right)$$

$$\cdot \left( \beta \cdot \mathbb{1}[x_k^{(i)} = \texttt{Lit}(n)] + (1 - \beta) \cdot \mathbb{1}[x_k^{(i)} = \neg\texttt{Lit}(n)] \right)$$

$$= \sum_{\boldsymbol{x} \in \{\boldsymbol{x} : \boldsymbol{x} \in \texttt{val}(\mathbf{X}) \wedge x_k = \texttt{Lit}(n)\}} \prod_{l=1, l \neq k}^{K} \left( \beta \cdot \mathbb{1}[x_l^{(i)} = x_l] + (1-\beta) \cdot \mathbb{1}[x_l^{(i)} \neq x_l] \right)$$

$$\cdot \left( \beta \cdot \mathbb{1}[x_k^{(i)} = x_k] \cdot \mathbb{1}[x_k = \texttt{Lit}(n)] + (1 - \beta) \cdot \mathbb{1}[x_k^{(i)} \neq x_k] \cdot \mathbb{1}[x_k = \texttt{Lit}(n)] \right)$$

$$= \sum_{\boldsymbol{x} \in \{\boldsymbol{x} : \boldsymbol{x} \in \texttt{val}(\mathbf{X}) \wedge x_k = \texttt{Lit}(n)\}} \prod_{l=1, l \neq k}^{K} \left( \beta \cdot \mathbb{1}[x_l^{(i)} = x_l] + (1-\beta) \cdot \mathbb{1}[x_l^{(i)} \neq x_l] \right)$$

$$\cdot \left( \beta \cdot \mathbb{1}[x_k^{(i)} = x_k] + (1 - \beta) \cdot \mathbb{1}[x_k^{(i)} \neq x_k] \right) \cdot \mathbb{1}[x_k = \texttt{Lit}(n)]$$

$$\overset{(b)}{=} \sum_{\boldsymbol{x} \in \{\boldsymbol{x} : \boldsymbol{x} \in \texttt{val}(\mathbf{X})\}} \prod_{l=1}^{K} \left( \beta \cdot \mathbb{1}[x_l^{(i)} = x_l] + (1-\beta) \cdot \mathbb{1}[x_l^{(i)} \neq x_l] \right) \cdot \mathbb{1}[x_k = \texttt{Lit}(n)]$$

$$= \sum_{\boldsymbol{x}\in\{\boldsymbol{x}:\boldsymbol{x}\in\mathsf{val}(\mathbf{X})\}}\prod_{l=1}^{K}\left(\beta\cdot\mathbb{1}[x_l^{(i)}=x_l]+(1-\beta)\cdot\mathbb{1}[x_l^{(i)}\neq x_l]\right)\cdot\mathbb{1}[\boldsymbol{x}\in\mathsf{supp}(n)],$$

where $(a)$ holds because the added term

$$\sum_{\boldsymbol{x}\in\{\boldsymbol{x}:\boldsymbol{x}\in\mathsf{val}(\mathbf{X})\wedge x_k=\mathtt{Lit}(n)\}}\prod_{l=1,l\neq k}^{K}\left(\beta\cdot\mathbb{1}[x_l^{(i)}=x_l]+(1-\beta)\cdot\mathbb{1}[x_l^{(i)}\neq x_l]\right)=1;$$

the sum condition $\boldsymbol{x}_k=\mathtt{Lit}(n)$ after $(b)$ can be lifted thanks to the indicator $\mathbb{1}[x_k=\mathtt{Lit}(n)]$.

• Inductive case: product units. Suppose $n$ is a product unit with children $\{c_j\}_{j=1}^{|\mathsf{in}(n)|}$. Recall that the scope of the child $c_j$ is denoted as $\phi(c_j)$. Since the PC is decomposable, the contexts of different children are non-overlapping. Suppose the value of any child unit $c_j$ is defined according to Eq. (6), i.e.,

$$\mathtt{value}_i[c_j]=\sum_{\boldsymbol{x}\in\mathsf{val}(\mathbf{X})}\prod_{k=1}^{K}\left(\beta\cdot\mathbb{1}[x_k^{(i)}=x_k]+(1-\beta)\cdot\mathbb{1}[x_k^{(i)}\neq x_k]\right)\cdot\mathbb{1}[\boldsymbol{x}\in\mathsf{supp}(c_j)].$$

Denote $K_{c_j}$ as the set of index for the variables in $\phi(c_j)$. We have

$$\mathtt{value}_i[n]\overset{(a)}{=}\prod_{j=1}^{|\mathsf{in}(n)|}\mathtt{value}_i[c_j]$$

$$=\prod_{j=1}^{|\mathsf{in}(n)|}\left\{\sum_{\boldsymbol{x}\in\mathsf{val}(\mathbf{X})}\prod_{k=1}^{K}\left(\beta\cdot\mathbb{1}[x_k^{(i)}=x_k]+(1-\beta)\cdot\mathbb{1}[x_k^{(i)}\neq x_k]\right)\cdot\mathbb{1}[\boldsymbol{x}\in\mathsf{supp}(c_j)]\right\}$$

$$=\prod_{j=1}^{|\mathsf{in}(n)|}\left\{\sum_{\boldsymbol{x}\in\mathsf{val}(\phi(c_j))}\prod_{k\in K_{c_j}}\left(\beta\cdot\mathbb{1}[x_k^{(i)}=x_k]+(1-\beta)\cdot\mathbb{1}[x_k^{(i)}\neq x_k]\right)\cdot\mathbb{1}[\boldsymbol{x}\in\mathsf{supp}(c_j)]\right\}$$

$$\overset{(b)}{=}\sum_{\boldsymbol{x}\in\mathsf{val}(\bigcup_{j=1}^{|\mathsf{in}(n)|}\phi(c_j))}\prod_{k\in\bigcup_{j=1}^{|\mathsf{in}(n)|}K_{c_j}}\left(\beta\cdot\mathbb{1}[x_k^{(i)}=x_k]+(1-\beta)\cdot\mathbb{1}[x_k^{(i)}\neq x_k]\right)$$

$$\cdot\left(\prod_{l=1}^{|\mathsf{in}(n)|}\mathbb{1}[\boldsymbol{x}\in\mathsf{supp}(c_l)]\right)$$

$$\overset{(c)}{=}\sum_{\boldsymbol{x}\in\mathsf{val}(\bigcup_{j=1}^{|\mathsf{in}(n)|}\phi(c_j))}\prod_{k\in\bigcup_{j=1}^{|\mathsf{in}(n)|}K_{c_j}}\left(\beta\cdot\mathbb{1}[x_k^{(i)}=x_k]+(1-\beta)\cdot\mathbb{1}[x_k^{(i)}\neq x_k]\right)\mathbb{1}[\boldsymbol{x}\in\mathsf{supp}(n)]$$

$$\overset{(d)}{=}\sum_{\boldsymbol{x}\in\mathsf{val}(\mathbf{X})}\prod_{k=1}^{K}\left(\beta\cdot\mathbb{1}[x_k^{(i)}=x_k]+(1-\beta)\cdot\mathbb{1}[x_k^{(i)}\neq x_k]\right)\mathbb{1}[\boldsymbol{x}\in\mathsf{supp}(n)],$$

where $(a)$ holds by line 6 of Alg. 1; $(b)$ holds since $\forall c_i, c_j\in\mathsf{in}(n)(c_i\neq c_j)$, we have $\phi(c_i)\cap\phi(c_j)=\varnothing$ and $K_{c_i}\cap K_{c_j}=\varnothing$ thanks to decomposability of the PC; $(c)$ is satisfied by the definition of product units: $\mathsf{supp}(n)=\bigcap_{c\in\mathsf{in}(n)}\mathsf{supp}(c)$; $(d)$ holds since $\bigcup_{j=1}^{|\mathsf{in}(n)|}\phi(c_j)$ is a subset of $\mathbf{X}$.

• Inductive case: sum units. Suppose $n$ is a sum unit with children $\{c_j\}_{j=1}^{|\mathsf{in}(n)|}$. Suppose the value $\mathtt{value}_i[c_j]$ of any child unit $c_j$ is defined according to Eq. (6), we have

$$\mathtt{value}_i[n]\overset{(a)}{=}\sum_{j=1}^{|\mathsf{in}(n)|}\mathtt{value}_i[c_j]$$

$$=\sum_{j=1}^{|\mathsf{in}(n)|}\left\{\sum_{\boldsymbol{x}\in\mathsf{val}(\mathbf{X})}\prod_{k=1}^{K}\left(\beta\cdot\mathbb{1}[x_k^{(i)}=x_k]+(1-\beta)\cdot\mathbb{1}[x_k^{(i)}\neq x_k]\right)\cdot\mathbb{1}[\boldsymbol{x}\in\mathsf{supp}(c_j)]\right\}$$

$$\overset{(b)}{=} \sum_{\boldsymbol{x}\in\mathsf{val}(\mathbf{X})} \prod_{k=1}^{K} \left(\beta\cdot\mathbb{1}[x_k^{(i)} = x_k] + (1-\beta)\cdot\mathbb{1}[x_k^{(i)} \neq x_k]\right) \cdot \left(\sum_{j=1}^{|\mathsf{in}(n)|} \mathbb{1}[\boldsymbol{x}\in\mathsf{supp}(c_j)]\right)$$

$$\overset{(c)}{=} \sum_{\boldsymbol{x}\in\mathsf{val}(\mathbf{X})} \prod_{k=1}^{K} \left(\beta\cdot\mathbb{1}[x_k^{(i)} = x_k] + (1-\beta)\cdot\mathbb{1}[x_k^{(i)} \neq x_k]\right) \cdot \mathbb{1}[\boldsymbol{x}\in\mathsf{supp}(n)],$$

where $(a)$ follows line 8 of Alg. 1; $(b)$ holds because the sum unit $n$ is deterministic: $\forall c_i, c_j \in \mathsf{in}(n)(c_i \neq c_j), \mathsf{supp}(c_i) \cap \mathsf{supp}(c_j) = \varnothing$; $(c)$ follows from the definition of sum units: $\mathsf{supp}(n) = \bigcup_{c\in\mathsf{in}(n)} \mathsf{supp}(c)$.

We have shown that for any unit $n$, the value stored in $\mathtt{value}_i[n]$ follows the definition in Eq. (6). We proceed to show the correctness of the backward pass.

**Correctness of the backward pass** Similar to the forward pass, we show that the context $\mathtt{context}_i[n]$ of each sum unit w.r.t. sample $\boldsymbol{x}^{(i)}$ computed by Alg. 2 is defined as

$$\mathtt{context}_i[n] = \sum_{\boldsymbol{x}\in\mathsf{val}(\mathbf{X})} \prod_{k=1}^{K} \left(\beta\cdot\mathbb{1}[x_k^{(i)} = x_k] + (1-\beta)\cdot\mathbb{1}[x_k^{(i)} \neq x_k]\right) \cdot \mathbb{1}[\boldsymbol{x}\in\gamma_n], \tag{7}$$

and the flow $\mathtt{flow}_i[n, c]$ of each edge $(n, c)$ s.t. $n$ is a sum unit is:

$$\mathtt{flow}_i[n, c] = \sum_{\boldsymbol{x}\in\mathsf{val}(\mathbf{X})} \prod_{k=1}^{K} \left(\beta\cdot\mathbb{1}[x_k^{(i)} = x_k] + (1-\beta)\cdot\mathbb{1}[x_k^{(i)} \neq x_k]\right) \cdot \mathbb{1}[\boldsymbol{x}\in\gamma_n \wedge \boldsymbol{x}\in\gamma_c]. \tag{8}$$

• Base case: root unit $n_r$. Without loss of generality, we assume the root node represents a sum unit.[6] According to Def. 5, the context of the root node $n_r$ equals its support, i.e., $\gamma_{n_r} = \mathsf{supp}(n_r)$. Since in line 3 of Alg. 2, the value $\mathtt{context}_i[n]$ is set to $\mathtt{value}_i[n]$, we know that

$$\mathtt{context}_i[n] = \sum_{\boldsymbol{x}\in\mathsf{val}(\mathbf{X})} \prod_{k=1}^{K} \left(\beta\cdot\mathbb{1}[x_k^{(i)} = x_k] + (1-\beta)\cdot\mathbb{1}[x_k^{(i)} \neq x_k]\right) \cdot \mathbb{1}[\boldsymbol{x}\in\mathsf{supp}(n)]$$

$$= \sum_{\boldsymbol{x}\in\mathsf{val}(\mathbf{X})} \prod_{k=1}^{K} \left(\beta\cdot\mathbb{1}[x_k^{(i)} = x_k] + (1-\beta)\cdot\mathbb{1}[x_k^{(i)} \neq x_k]\right) \cdot \mathbb{1}[\boldsymbol{x}\in\gamma_n].$$

• Inductive case: sum unit. Suppose $n$ is a sum unit with parent product units $\{m_j\}_{j=1}^{|\mathsf{pa}(n)|}$. Denote the parent of product unit $m_i$ as $g_i$.[7] Suppose the contexts of $\{g_j\}_{j=1}^{|\mathsf{pa}(n)|}$ satisfy Eq. (7). For ease of presentation, denote $H(\boldsymbol{x}, \boldsymbol{x}^{(i)}, k) := \left(\beta\cdot\mathbb{1}[x_k^{(i)} = x_k] + (1-\beta)\cdot\mathbb{1}[x_k^{(i)} \neq x_k]\right)$.

$$\mathtt{flow}_i[g_j, m_j] = \frac{\mathtt{value}_i[m_j]}{\mathtt{value}_i[g_j]} \cdot \mathtt{context}_i[g_j]$$

$$= \frac{\sum_{\boldsymbol{x}\in\mathsf{val}(\mathbf{X})} \prod_{k=1}^{K} H(\boldsymbol{x}, \boldsymbol{x}^{(i)}, k) \cdot \mathbb{1}[\boldsymbol{x}\in\gamma_{g_j}]}{\sum_{\boldsymbol{x}\in\mathsf{val}(\mathbf{X})} \prod_{k=1}^{K} H(\boldsymbol{x}, \boldsymbol{x}^{(i)}, k) \cdot \mathbb{1}[\boldsymbol{x}\in\mathsf{supp}(g_j)]}$$

$$\cdot \sum_{\boldsymbol{x}\in\mathsf{val}(\mathbf{X})} \prod_{k=1}^{K} H(\boldsymbol{x}, \boldsymbol{x}^{(i)}, k) \cdot \mathbb{1}[\boldsymbol{x}\in\mathsf{supp}(m_j)] \tag{9}$$

Define $\gamma'_{g_j} := \bigcup_{c\in\mathsf{pa}(g_j)} \gamma_c$, Def. 5 suggests that $\gamma_{g_j} = \gamma'_{g_j} \cap \mathsf{supp}(g_j)$. Thus,

$$\mathbb{1}[\boldsymbol{x}\in\gamma_{g_j}] = \mathbb{1}[\boldsymbol{x}\in\gamma'_{g_j}] \cdot \mathbb{1}[\boldsymbol{x}\in\mathsf{supp}(g_j)]. \tag{10}$$

---

[6]Note that if the root unit is not a sum, we can always add a sum unit as its parent and set the corresponding edge parameter to 1.

[7]W.l.o.g. we assume all product unit only have one parent.

Consider conditioning $\mathsf{supp}(g_j)$ and $\gamma'_{g_j}$ on the variables $\phi(g_j)$ (i.e., the variable scope of $g_j$). For any partial variable assignment $\boldsymbol{e}$ over $\phi(g_j)$, if $\boldsymbol{e} \in \mathsf{supp}(g_j)$, then $\boldsymbol{e} \in \gamma'_{g_j}$. Denote $K_{g_j}$ as the set of index for the variables in $\phi(g_j)$. We have

$$\sum_{\boldsymbol{x} \in \mathsf{val}(\mathbf{X})} \prod_{k=1}^{K} H(\boldsymbol{x}, \boldsymbol{x}^{(i)}, k) \cdot \mathbb{1}[\boldsymbol{x} \in \gamma'_{g_j}] \cdot \mathbb{1}[\boldsymbol{x} \in \mathsf{supp}(g_j)]$$

$$= \left( \sum_{\boldsymbol{x} \in \mathsf{val}(\phi(g_j))} \prod_{k \in K_{g_j}} H(\boldsymbol{x}, \boldsymbol{x}^{(i)}, k) \cdot \mathbb{1}[\boldsymbol{x} \in \mathsf{supp}(g_j)] \right)$$

$$\cdot \left( \sum_{\boldsymbol{x} \in \mathsf{val}(\mathbf{X} \setminus \phi(g_j))} \prod_{k \in \{1,...,K\} \setminus K_{g_j}} H(\boldsymbol{x}, \boldsymbol{x}^{(i)}, k) \cdot \mathbb{1}[\boldsymbol{x} \in \gamma'_{g_j}] \right) \tag{11}$$

Plug Eqs. (11) and (10) into Eq. (9), we have

$$\mathtt{flow}_i[g_j, m_j] = \frac{\sum_{\boldsymbol{x} \in \mathsf{val}(\phi(g_j))} \prod_{k \in K_{g_j}} H(\boldsymbol{x}, \boldsymbol{x}^{(i)}, k) \cdot \mathbb{1}[\boldsymbol{x} \in \mathsf{supp}(g_j)]}{\sum_{\boldsymbol{x} \in \mathsf{val}(\mathbf{X})} \prod_{k=1}^{K} H(\boldsymbol{x}, \boldsymbol{x}^{(i)}, k) \cdot \mathbb{1}[\boldsymbol{x} \in \mathsf{supp}(g_j)]}$$

$$\cdot \left( \sum_{\boldsymbol{x} \in \mathsf{val}(\mathbf{X} \setminus \phi(g_j))} \prod_{k \in \{1,...,K\} \setminus K_{g_j}} H(\boldsymbol{x}, \boldsymbol{x}^{(i)}, k) \cdot \mathbb{1}[\boldsymbol{x} \in \gamma'_{g_j}] \right)$$

$$\cdot \left( \sum_{\boldsymbol{x} \in \mathsf{val}(\mathbf{X})} \prod_{k=1}^{K} H(\boldsymbol{x}, \boldsymbol{x}^{(i)}, k) \cdot \mathbb{1}[\boldsymbol{x} \in \mathsf{supp}(m_j)] \right)$$

$$= \left( \sum_{\boldsymbol{x} \in \mathsf{val}(\mathbf{X} \setminus \phi(g_j))} \prod_{k \in \{1,...,K\} \setminus K_{g_j}} H(\boldsymbol{x}, \boldsymbol{x}^{(i)}, k) \cdot \mathbb{1}[\boldsymbol{x} \in \gamma'_{g_j}] \right)$$

$$\cdot \left( \sum_{\boldsymbol{x} \in \mathsf{val}(\phi(g_j))} \prod_{k \in K_{g_j}} H(\boldsymbol{x}, \boldsymbol{x}^{(i)}, k) \cdot \mathbb{1}[\boldsymbol{x} \in \mathsf{supp}(m_j)] \right) \tag{12}$$

Since $m_j$ is a child of $g_j$, the support of $m_j$ is a subset of $g_j$'s support: $\mathsf{supp}(m_j) \subseteq \mathsf{supp}(g_j)$. Therefore, for any partial variable assignment $\boldsymbol{e}$ over $\phi(g_j)$, if $\boldsymbol{e} \in \mathsf{supp}(m_j)$, then $\boldsymbol{e} \in \mathsf{supp}(g_j)$. Since $\{\boldsymbol{e} \mid \boldsymbol{e} \in \mathsf{val}(\phi(g_j)) \wedge \boldsymbol{e} \in \mathsf{supp}(g_j)\} \subseteq \{\boldsymbol{e} \mid \boldsymbol{e} \in \mathsf{val}(\phi(g_j)) \wedge \boldsymbol{e} \in \gamma'_{g_j}\}$, we conclude that for any partial variable assignment $\boldsymbol{e}$ over $\phi(g_j)$, if $\boldsymbol{e} \in \mathsf{supp}(m_j)$, then $\boldsymbol{e} \in \gamma'_{g_j}$. Therefore, the two product terms in Eq. (12) can be joined with a Cartesian product:

$$\mathtt{flow}_i[g_j, m_j] = \sum_{\boldsymbol{x} \in \mathsf{val}(\mathbf{X})} \prod_{k=1}^{K} H(\boldsymbol{x}, \boldsymbol{x}^{(i)}, k) \cdot \mathbb{1}[\boldsymbol{x} \in \gamma'_{g_j} \cap \mathsf{supp}(m_j)]. \tag{13}$$

Note that $\gamma'_{g_j} \cap \mathsf{supp}(m_j) = \overline{\gamma'_{g_j} \cup \mathsf{supp}(g_j)} \cap \mathsf{supp}(m_j) = \gamma_{g_j} \cap \mathsf{supp}(m_j)$. Since $\gamma_{m_j} = \gamma_{g_j} \cap \mathsf{supp}(m_j)$ (according to Def. 5), we have

$$\gamma'_{g_j} \cap \mathsf{supp}(m_j) = \gamma_{g_j} \cap \mathsf{supp}(m_j)$$

$$= \gamma_{g_j} \cap \mathsf{supp}(g_j) \cap \mathsf{supp}(m_j)$$

$$= \gamma_{g_j} \cap \gamma_{m_j}.$$

Plug the above equation into Eq. (13), we have

$$\mathtt{flow}_i[g_j, m_j] = \sum_{\boldsymbol{x} \in \mathsf{val}(\mathbf{X})} \prod_{k=1}^{K} H(\boldsymbol{x}, \boldsymbol{x}^{(i)}, k) \cdot \mathbb{1}[\boldsymbol{x} \in \gamma_{g_j} \cap \gamma_{m_j}],$$

which is equivalent to Eq. (7).

We proceed to show that the context of unit $n$ follows Eq. (8). According to lines 6 and 7 of Alg. 2, $\mathtt{context}_i[n]$ is computed as

$$\mathtt{context}_i[n] = \sum_{j=1}^{|\mathsf{pa}(n)|} \mathtt{flow}_i[g_j, m_j]$$

$$= \sum_{j=1}^{|\mathsf{pa}(n)|} \sum_{\boldsymbol{x} \in \mathsf{val}(\mathbf{X})} \prod_{k=1}^{K} H(\boldsymbol{x}, \boldsymbol{x}^{(i)}, k) \cdot \mathbb{1}[\boldsymbol{x} \in \gamma_{g_j} \cap \gamma_{m_j}]$$

$$= \sum_{\boldsymbol{x} \in \mathsf{val}(\mathbf{X})} \prod_{k=1}^{K} H(\boldsymbol{x}, \boldsymbol{x}^{(i)}, k) \cdot \Big( \sum_{j=1}^{|\mathsf{pa}(n)|} \mathbb{1}[\boldsymbol{x} \in \gamma_{g_j} \cap \gamma_{m_j}] \Big). \tag{14}$$

Next, we show that $\forall m_i, m_j \in \mathsf{pa}(n)(m_i \neq m_j), \gamma_{m_i} \cap \gamma_{m_j} = \varnothing$. We prove this claim using its contrapositive form. Suppose there exists $\boldsymbol{x} \in \mathsf{val}(\mathbf{X})$ such that $\boldsymbol{x} \in \gamma_{m_i}$ and $\boldsymbol{x} \in \gamma_{m_j}$. According to the definition of context, if $\boldsymbol{x} \in \gamma_{m_i}$, then there must be a path between $m_i$ and the root node $n_r$ where all nodes in the path are "activated", i.e., for any unit $c$ in the path, $\boldsymbol{x} \in \gamma_c$. Similarly, there much exists a path of "activated" units between $m_j$ and $n_r$. We note that the two paths must share a set of identical nodes since their terminal are both the root node $n_r$. Therefore, there must exist a sum unit $n'$ along the intersection of the two path where at least two of its children are activated, i.e., $\exists c_1, c_2 \in \mathsf{in}(n')(c_1 \neq c_2)$, such that $\boldsymbol{x} \in \gamma_{c_1}$ and $\boldsymbol{x} \in \gamma_{c_2}$. This contradicts the assumption that the PC is deterministic. Therefore, the claim at the beginning of this paragraph holds. Thus,

$$\sum_{j=1}^{|\mathsf{pa}(n)|} \mathbb{1}[\boldsymbol{x} \in \gamma_{g_j} \cap \gamma_{m_j}] \overset{(a)}{=} \sum_{j=1}^{|\mathsf{pa}(n)|} \mathbb{1}[\boldsymbol{x} \in \gamma_{m_j}]$$

$$\overset{(b)}{=} \mathbb{1}[\boldsymbol{x} \in \bigcup_{j=1}^{|\mathsf{pa}(n)|} \gamma_{m_j}]$$

$$\overset{(c)}{=} \mathbb{1}[\boldsymbol{x} \in \bigcup_{j=1}^{|\mathsf{pa}(n)|} \gamma_{m_j} \cap \mathsf{supp}(n)]$$

$$\overset{(d)}{=} \mathbb{1}[\boldsymbol{x} \in \gamma_n],$$

where $(a)$ follows from $\gamma_{m_j} \subseteq \gamma_{g_j}$; $(b)$ holds because the statement made in the previous paragraph (i.e., $\forall m_i, m_j \in \mathsf{pa}(n)(m_i \neq m_j), \gamma_{m_i} \cap \gamma_{m_j} = \varnothing$); $(c)$ holds since $\mathsf{supp}(m_j) \subseteq \mathsf{supp}(n)$ and $\gamma_{m_j} \subseteq \mathsf{supp}(m_j)$; $(d)$ directly applies the definition of context (i.e., Def. 5).

Plug in Eq. (14), we have

$$\mathtt{context}_i[n] = \sum_{\boldsymbol{x} \in \mathsf{val}(\mathbf{X})} \prod_{k=1}^{K} H(\boldsymbol{x}, \boldsymbol{x}^{(i)}, k) \cdot \Big( \sum_{j=1}^{|\mathsf{pa}(n)|} \mathbb{1}[\boldsymbol{x} \in \gamma_{g_j} \cap \gamma_{m_j}] \Big)$$

$$= \sum_{\boldsymbol{x} \in \mathsf{val}(\mathbf{X})} \prod_{k=1}^{K} H(\boldsymbol{x}, \boldsymbol{x}^{(i)}, k) \cdot \mathbb{1}[\boldsymbol{x} \in \gamma_n].$$

**Computing $\mathrm{F}_{n,c}(\mathcal{D}_\beta)$** Finally, we can compute $\mathrm{F}_{n,c}(\mathcal{D}_\beta)$ from the flows (i.e., $\mathtt{flow}_i[n,c]$) computed by Alg. 2:

$$\mathrm{F}_{n,c}(\mathcal{D}_\beta) = \sum_{\boldsymbol{x} \in \mathsf{val}(\mathbf{X})} \mathtt{weight}(\mathcal{D}_\beta, \boldsymbol{x}) \cdot \mathbb{1}[\boldsymbol{x} \in \gamma_n \wedge \boldsymbol{x} \in \gamma_c]$$

$$= \sum_{\boldsymbol{x} \in \mathsf{val}(\mathbf{X})} \underbrace{\sum_{i=1}^{N} \prod_{k=1}^{K} \Big( \beta \cdot \mathbb{1}[x_k^{(i)} = x_k] + (1-\beta) \cdot \mathbb{1}[x_k^{(i)} \neq x_k] \Big)}_{\mathtt{weight}(\mathcal{D}_\beta, \boldsymbol{x})} \cdot \mathbb{1}[\boldsymbol{x} \in \gamma_n \wedge \boldsymbol{x} \in \gamma_c]$$

$$= \sum_{i=1}^{N} \underbrace{\sum_{\boldsymbol{x} \in \mathsf{val}(\mathbf{X})} \prod_{k=1}^{K} \Big( \beta \cdot \mathbb{1}[x_k^{(i)} = x_k] + (1-\beta) \cdot \mathbb{1}[x_k^{(i)} \neq x_k] \Big) \cdot \mathbb{1}[\boldsymbol{x} \in \gamma_n \wedge \boldsymbol{x} \in \gamma_c]}_{\mathtt{flow}_i[n,c]}$$

$$= \sum_{i=1}^{N} \mathtt{flow}_i[n,c].$$

Finally, we note that Alg. 1 and 2 both run in time $\mathcal{O}(|p| \cdot |\mathcal{D}|)$.

---
**Algorithm 4** PC entropy
---
1: **Input:** A deterministic PC $p$
2: **Output:** $\texttt{entropy}[n] := \texttt{ENT}(p_n)$ for every unit $n$
3: **foreach** $n$ traversed in postorder **do**
4:     **if** $n$ **isa** input unit **then** $\texttt{entropy}[n] = \texttt{ENT}(p_n)$ //entropy of the input distribution
5:     **elif** $n$ **isa** product unit **then** $\texttt{entropy}[n] = \sum_{c \in \text{in}(n)} \texttt{entropy}[c]$
6:     **else** //$n$ is a sum unit **then**

$$\texttt{entropy}[n] = -\sum_{c \in \text{in}(n)} \theta_{n,c} \log \theta_{n,c} + \sum_{c \in \text{in}(n)} \theta_{n,c} \cdot \texttt{entropy}[c]$$

---

## A.2 Useful Lemmas

This section provides several useful lemmas that are later used in the proof of Thm. 4.

**Lemma 1.** *Given a deterministic PC $p$ whose root node is $n_r$, its entropy $\texttt{ENT}(p) \stackrel{def}{=} \texttt{ENT}(p_{n_r})$ can be decomposed recursively as follows:*

$$\texttt{ENT}(p_n) = \begin{cases} \sum_{c \in \text{in}(n)} \big( -\theta_{n,c} \log \theta_{n,c} + \theta_{n,c} \cdot \texttt{ENT}(p_c) \big) & \text{if } n \text{ is a sum unit,} \\ \sum_{c \in \text{in}(n)} \texttt{ENT}(p_c) & \text{if } n \text{ is a product unit,} \end{cases}$$

*where the entropy of an input unit is defined by the entropy of the corresponding univariate distribution. Following this decomposition, we construct Alg. 4 that computes the entropy of every nodes in a deterministic PC in $\mathcal{O}(|p|)$ time.*

*Proof.* We show the correctness of the entropy decomposition over a sum unit and a product unit respectively.

● Sum units. If $n$ is a sum unit:

$$\texttt{ENT}(p_n) = -\sum_{\boldsymbol{x} \in \text{val}(\phi(n))} \Big( \sum_{c \in \text{in}(n)} \theta_{n,c} p_c(\boldsymbol{x}) \Big) \log \Big( \sum_{c \in \text{in}(n)} \theta_{n,c} p_c(\boldsymbol{x}) \Big)$$

$$= -\sum_{\boldsymbol{x} \in \text{val}(\phi(n))} \Big( \sum_{c \in \text{in}(n)} \theta_{n,c} p_c(\boldsymbol{x}) \mathbb{1}[\boldsymbol{x} \in \text{supp}(c)] \Big) \log \Big( \sum_{c \in \text{in}(n)} \theta_{n,c} p_c(\boldsymbol{x}) \mathbb{1}[\boldsymbol{x} \in \text{supp}(c)] \Big)$$

$$\stackrel{(a)}{=} -\sum_{\boldsymbol{x} \in \text{val}(\phi(n))} \sum_{c \in \text{in}(n)} \mathbb{1}[\boldsymbol{x} \in \text{supp}(c)] \cdot \theta_{n,c} \cdot p_c(\boldsymbol{x}) \cdot \Big( \log \theta_{n,c} + \log p_c(\boldsymbol{x}) \Big)$$

$$= -\sum_{c \in \text{in}(n)} \theta_{n,c} \log \theta_{n,c} \underbrace{\Big( \sum_{\boldsymbol{x} \in \text{val}(\phi(n))} \mathbb{1}[\boldsymbol{x} \in \text{supp}(c)] p_c(\boldsymbol{x}) \Big)}_{=1}$$

$$+ \sum_{c \in \text{in}(n)} \theta_{n,c} \underbrace{\Big( -\sum_{\boldsymbol{x} \in \text{val}(\phi(n))} p_c(\boldsymbol{x}) \log p_c(\boldsymbol{x}) \Big)}_{=\texttt{ENT}(p_c)}$$

$$= \sum_{c \in \text{in}(n)} \Big( -\theta_{n,c} \log \theta_{n,c} + \theta_{n,c} \cdot \texttt{ENT}(p_c) \Big), \tag{15}$$

where $(a)$ uses the assumption that the sum unit is deterministic, i.e., $\forall c_1, c_2 \in \text{in}(n)$ $(c_1 \neq c_2)$, $\text{supp}(c_1) \cap \text{supp}(c_2) = \varnothing$.

● Product units. If $n$ is a product unit:

$$\texttt{ENT}(p_n) = -\sum_{\boldsymbol{x} \in \text{val}(\phi(n))} \Big( \prod_{c \in \text{in}(n)} p_c(\boldsymbol{x}) \Big) \log \Big( \prod_{c \in \text{in}(n)} p_c(\boldsymbol{x}) \Big)$$

$$= -\sum_{c \in \text{in}(n)} \Big( \sum_{\boldsymbol{x} \in \text{val}(\phi(c))} p_c(\boldsymbol{x}) \log p_c(\boldsymbol{x}) \Big)$$

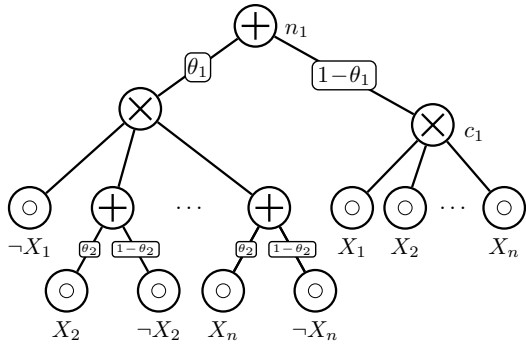

Figure 7: An example PC to show that PC entropy is neither convex nor concave.

$$= \sum_{c \in \mathsf{in}(n)} \mathrm{ENT}(p_c). \tag{16}$$

$\square$

**Lemma 2.** *The entropy of a deterministic PC $p$ is neither convex nor concave w.r.t. its parameters.*

*Proof.* Consider the example PC in Fig. 7. Assume $n = 20$ and define parameters $\boldsymbol{\theta}_a = \{\theta_1 = 0.1, \theta_2 = 0.1\}$ and $\boldsymbol{\theta}_b = \{\theta_1 = 0.12, \theta_2 = 0.12\}$. Denote $\boldsymbol{\theta}_c = (\boldsymbol{\theta}_a + \boldsymbol{\theta}_b)/2$, we have

$$2 \cdot \mathrm{ENT}(p; \boldsymbol{\theta}_c) - \mathrm{ENT}(p; \boldsymbol{\theta}_a) - \mathrm{ENT}(p; \boldsymbol{\theta}_b) \approx -0.0047898 < 0.$$

Hence the entropy is not concave.

Define parameters $\boldsymbol{\theta}_d = \{\theta_1 = 0.4, \theta_2 = 0.8\}$ and $\boldsymbol{\theta}_e = \{\theta_1 = 0.42, \theta_2 = 0.82\}$. Denote $\boldsymbol{\theta}_f = (\boldsymbol{\theta}_d + \boldsymbol{\theta}_e)/2$, we have

$$2 \cdot \mathrm{ENT}(p; \boldsymbol{\theta}_f) - \mathrm{ENT}(p; \boldsymbol{\theta}_d) - \mathrm{ENT}(p; \boldsymbol{\theta}_e) \approx 0.0056294 > 0.$$

Hence the entropy is not convex. $\square$

**Lemma 3.** *For any dataset $\mathcal{D} = \{\boldsymbol{x}^{(i)}\}_{i=1}^N$ and any deterministic PC $p$ with parameters $\boldsymbol{\theta}$, the following formula is concave w.r.t. $\boldsymbol{\theta}$:*

$$\sum_{i=1}^N \log p(\boldsymbol{x}^{(i)}; \boldsymbol{\theta}). \tag{17}$$

*Proof.* For any input $\boldsymbol{x}$, $\log p(\boldsymbol{x}; \boldsymbol{\theta})$ can be decomposed over sum and product units:

• Sum units. Suppose $n$ is a sum unit, then

$$\begin{aligned}
\log p_n(\boldsymbol{x}; \boldsymbol{\theta}) &= \log \Big( \sum_{c \in \mathsf{in}(n)} \theta_{n,c} \cdot p_c(\boldsymbol{x}) \Big) \\
&= \log \Big( \sum_{c \in \mathsf{in}(n)} \theta_{n,c} \cdot p_c(\boldsymbol{x}) \mathbb{1}[\boldsymbol{x} \in \mathsf{supp}(c)] \Big) \\
&= \sum_{c \in \mathsf{in}(n)} \mathbb{1}[\boldsymbol{x} \in \mathsf{supp}(c)] \big( \log \theta_{n,c} + \log p_c(\boldsymbol{x}) \big),
\end{aligned} \tag{18}$$

where the last equation holds because unit $n$ is deterministic: $\forall c_i, c_j \in \mathsf{in}(n)(c_i \neq c_j), \mathsf{supp}(c_i) \cap \mathsf{supp}(c_j) = \varnothing$.

• Product units. Suppose $n$ is a product unit, then

$$\log p_n(\boldsymbol{x}; \boldsymbol{\theta}) = \log \Big( \prod_{c \in \mathsf{in}(n)} p_c(\boldsymbol{x}) \Big) = \sum_{c \in \mathsf{in}(n)} \log p_c(\boldsymbol{x}). \tag{19}$$

According to Eqs. (18) and (19), for any $\boldsymbol{x} \in \mathsf{val}(\mathbf{X})$, $\log p(\boldsymbol{x}; \boldsymbol{\theta})$ can be decomposed into the sum over a set of log-parameters (e.g., $\log \theta_{n,c}$). Therefore, Eq. (17) is concave. $\square$

**Lemma 4.** *Given a deterministic PC $p$ with root node $n_r$, its entropy $\text{ENT}(p)$ can be decomposed as follows:*

$$\text{ENT}(p_{n_r}) = - \sum_{(n,c)\in\text{edges}(p_{n_r})} P_{n_r}(n) \cdot \theta_{n,c} \log \theta_{n,c},$$

*where $\text{edges}(p)$ denotes all edges $(n,c)$ in the PC with sum unit $n$; $P_{n_r}(n)$ is defined in Eq. (21).*

*Proof.* We prove the lemma by induction.

• Base case. Suppose $m$ is a sum unit such that all its decendents are either input units or product unit. By definition, we have $P_m(m) = 1$, and $\text{edges}(p_m) = \{(m,c) \mid c \in \text{in}(m)\}$. Thus,

$$- \sum_{(n,c)\in\text{edges}(p_m)} P_m(n) \cdot \theta_{n,c} \log \theta_{n,c} = - \sum_{c\in\text{in}(m)} \theta_{m,c} \log \theta_{m,c} = \text{ENT}(p_m).$$

• Inductive case: product units. Suppose $m$ is a product unit such that for each of its children $c \in \text{in}(m)$, we have

$$\text{ENT}(p_c) = - \sum_{(n',c')\in\text{edges}(p_c)} P_c(n') \cdot \theta_{n',c'} \log \theta_{n',c'}.$$

Then by Lem. 1 we know that

$$\text{ENT}(p_m) = \sum_{c\in\text{in}(m)} \text{ENT}(p_m)$$

$$= - \sum_{c\in\text{in}(m)} \sum_{(n',c')\in\text{edges}(p_c)} P_c(n') \cdot \theta_{n',c'} \log \theta_{n',c'}$$

$$\overset{(a)}{=} - \sum_{c\in\text{in}(m)} \sum_{(n',c')\in\text{edges}(p_c)} P_m(n') \cdot \theta_{n',c'} \log \theta_{n',c'}$$

$$\overset{(b)}{=} - \sum_{(n',c')\in\text{edges}(p_m)} P_m(n') \cdot \theta_{n',c'} \log \theta_{n',c'},$$

where $(a)$ holds since for any sum unit $n'$, $P_c(n') = P_m(n')$, and $(b)$ follows from the fact that $\text{edges}(p_m) = \bigcup_{c\in\text{in}(m)} \text{edges}(p_c)$.

• Inductive case: sum units. Suppose $m$ is a sum unit such that for each of its children $c \in \text{in}(m)$, we have

$$\text{ENT}(p_c) = - \sum_{(n',c')\in\text{edges}(p_c)} P_c(n') \cdot \theta_{n',c'} \log \theta_{n',c'}.$$

Then by Lem. 1 we have

$$\text{ENT}(p_m) = \sum_{c\in\text{in}(m)} \left( - \theta_{n,c} \log \theta_{n,c} + \theta_{n,c} \cdot \text{ENT}(p_c) \right)$$

$$= \sum_{c\in\text{in}(m)} -\theta_{n,c} \log \theta_{n,c} - \sum_{c\in\text{in}(n)} \sum_{(n',c')\in\text{edges}(p_c)} \underbrace{\theta_{m,c} \cdot P_c(n')}_{P_m(n')} \cdot \theta_{n',c'} \log \theta_{n',c'}$$

$$= \sum_{c\in\text{in}(m)} -P_m(m)\theta_{n,c} \log \theta_{n,c} - \sum_{c\in\text{in}(n)} \sum_{(n',c')\in\text{edges}(p_c)} P_m(n') \cdot \theta_{n',c'} \log \theta_{n',c'}$$

$$\overset{(a)}{=} - \sum_{(n',c')\in\text{edges}(p_m)} P_m(n') \cdot \theta_{n',c'} \log \theta_{n',c'},$$

where $(a)$ holds because $\text{edges}(p_m) = \left( \bigcup_{c\in\text{in}(m)} \text{edges}(p_c) \right) \bigcup \left( \{(m,c) \mid c \in \text{in}(m)\} \right)$. □

**Lemma 5.** *The entropy regularization objective in Eq. (2) w.r.t. a deterministic PC $p$ and a dataset $\mathcal{D}$ could have multiple local maximas.*

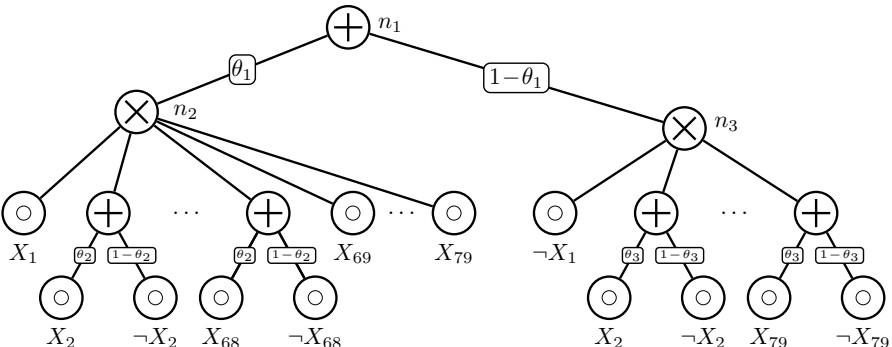

Figure 8: An example PC to show that Eq. (2) could have multiple stationary points.

*Proof.* Consider the deterministic PC $p$ in Fig. 8 and dataset $\mathcal{D}$ with a single sample $\boldsymbol{x} = (\texttt{true}, \ldots, \texttt{true})$. The objective in Eq. (2) can be re-written as follows

$$\mathcal{L}_{\mathrm{ent}}(\boldsymbol{\theta}; n_1, \tau) := \log p_{n_1}(\boldsymbol{x}) + \tau \cdot \mathrm{ENT}(p_{n_1}),$$

where $n_1$ is the root node of the PC as denoted in Fig. 8. We further decompose $\mathcal{L}_{\mathrm{ent}}(\boldsymbol{\theta}; n_1, \tau)$:

$$\begin{aligned}\mathcal{L}_{\mathrm{ent}}(\boldsymbol{\theta}; n_1, \tau) = {} & \log \theta_1 + 67 \cdot \log \theta_2 + \tau \cdot \mathrm{ENT}(p_{n_1}) \\ = {} & \log \theta_1 + 67 \cdot \log \theta_2 + \tau \cdot \left( -\theta_1 \log \theta_1 - (1-\theta_1) \log(1-\theta_1) \right) \\ & + \tau \cdot \theta_1 \cdot \mathrm{ENT}(p_{n_2}) + \tau \cdot (1-\theta_1) \cdot \mathrm{ENT}(p_{n_3}).\end{aligned}$$

First, we observe that to maximize $\mathcal{L}_{\mathrm{ent}}(\boldsymbol{\theta}; n_1, \tau)$, $\theta_3$ should always be $0.5$ since the only term that depends on $\theta_3$ is $(1-\theta_1) \cdot \mathrm{ENT}(p_{n_3})$ and $1-\theta_1 > 0$. Therefore, we have

$$\mathrm{ENT}(p_{n_3}) = 78 \cdot \log 2 \approx 54.065.$$

Next, for any fixed $\theta_1 \in (0, 1]$, the objective $\mathcal{L}_{\mathrm{ent}}(\boldsymbol{\theta}; n_1, \tau)$ is concave w.r.t. $\theta_2$:

$$\mathcal{L}_{\mathrm{ent}}(\boldsymbol{\theta}; n_1, \tau) = 67 \cdot \left( \log \theta_2 - \tau \cdot \theta_1 \cdot (\theta_2 \log \theta_2 + (1-\theta_2) \log(1-\theta_2)) \right) + \mathrm{const}, \qquad (20)$$

where the constant term does not depend on $\theta_2$. Therefore, for any $\theta_1$, we can uniquely compute the optimal value of $\theta_2$. We are left with determining the optimal value of $\theta_1$. Choose $\tau = 1.5$, the derivative of $\mathcal{L}_{\mathrm{ent}}(\boldsymbol{\theta}; n_1, \tau)$ w.r.t. $\theta_1$ is (denote $\texttt{ent}_0 := -(\theta_2 \log \theta_2 + (1-\theta_2) \log(1-\theta_2))$)

$$\begin{aligned}g(\theta_1) := \frac{\partial \mathcal{L}_{\mathrm{ent}}(\boldsymbol{\theta}; n_1, \tau)}{\partial \theta_1} & = \frac{1}{\theta_1} + 1.5 \cdot \left( \log(1-\theta_1) - \log \theta_1 + \mathrm{ENT}(p_{n_2}) - \mathrm{ENT}(p_{n_3}) \right) \\ & = \frac{1}{\theta_1} + 1.5 \cdot \left( \log(1-\theta_1) - \log \theta_1 + 67 \cdot \texttt{ent}_0 - \mathrm{ENT}(p_{n_3}) \right)\end{aligned}$$

where $\mathrm{ENT}(p_{n_3})$ can be viewed as a constant and $\texttt{ent}_0$ depends on $\theta_1$. Specifically, for any $\theta_1$, we compute $\theta_2$ and hence $\texttt{ent}_0$ by maximizing Eq. (20). Putting everything together, we have

$$\begin{cases} g(0.02) \approx 1.772730 > 0, \\ g(0.7) \approx -0.190743 < 0, \\ g(0.9) \approx 2.055231 > 0, \\ g(0.99) \approx -0.216938 < 0. \end{cases}$$

Since $g$ is continuous in range $(0, 1]$, there exists a local maxima of $\theta_1$ between $0.02$ and $0.7$ as well as between $0.9$ and $0.99$. Therefore, the entropy regularization objective could have multiple local maximas.

$\square$

## A.3  Proof of Theorem 1

This theorem is a direct corollary of Theorem 5 in [49], which has the following statement:

Computing the expectation of a logistic regression model w.r.t. a uniform data distribution is #-hard.

Note that with $\beta = 0.5$, the distribution $\mathcal{D}_\beta$ is essentially uniform, Thm. 1 follows directly from [49].

## A.4 Proof of Theorem 2

This proof largely follows the proof of Theorem 5 in [49]. The proof is by reduction from #NUMPAR, which is defined as follows. Given $n$ positive integers $k_1, \ldots, k_n$, we want to count the number of subset $S \subseteq [n]$ that satisfies $\sum_{i \in S} k_i = \sum_{i \notin S} k_i$. #NUMPAR is known to be #P-hard.

Fix an instance of #NUMPAR, $k_1, \ldots, k_n$, and assume w.l.o.g. that the sum of the numbers is even, i.e., $\sum_i k_i = 2c$ for some natural number $c$. Define $P := \{S \mid S \subseteq [n], \sum_{i \in S} k_i = c\}$. By definition $|P|$ is the solution to the #NUMPAR problem. Note that for each $S \in P$, its complement $\bar{S}$ should also be a member of $P$, and hence $|P|$ is even.

Define a logistic regression model as $F(x_1, \ldots, x_n) := \sigma(w_0 + \sum_{i=1}^n w_i \cdot x_i)$, where $\sigma$ is the sigmoid function. Define the normalized model of $F$ as $G(x_1, \ldots, x_n) := F(x_1, \ldots, x_n)/Z$, where $Z := \sum_{\boldsymbol{x} \in \mathsf{val}(X)} F(x_1, \ldots, x_n)$. Denote the entropy of a normalized logistic regressor $G$ as $\mathtt{ENT}(G) := -\sum_{\boldsymbol{x} \in \mathsf{val}(\mathbf{X})} G(\boldsymbol{x}) \log G(\boldsymbol{x})$.

We now describe an algorithm that computes $|P|$ using an oracle for $\mathtt{ENT}(G)$, where $G$ is a normalized logistic regression model. Denote $m$ as a large natural number to be chosen later, and define the following weights

$$w_0 := -\frac{m}{2} - mc, \quad w_i := mk_i (\forall i \in [n]).$$

Let $F$ be the logistic regressor corresponds to the above weights and $G$ the normalized model of $F$. We can represent $\mathtt{ENT}(G)$ as follows:

$$\mathtt{ENT}(G) = -\sum_{\boldsymbol{x} \in \mathsf{val}(\mathbf{X})} \frac{F(\boldsymbol{x})}{Z} \log \frac{F(\boldsymbol{x})}{Z} = -\sum_{\boldsymbol{x} \in \mathsf{val}(\mathbf{X})} \left( \frac{F(\boldsymbol{x}) \log F(\boldsymbol{x})}{Z} - \frac{F(\boldsymbol{x})}{Z} \log Z \right)$$

$$= -\sum_{\boldsymbol{x} \in \mathsf{val}(\mathbf{X})} \frac{F(\boldsymbol{x}) \log F(\boldsymbol{x})}{Z} + \log Z.$$

For large enough $m$, $F(\boldsymbol{x})$ will approach either $0$ or $1$. Therefore, the first term in the above equation will approach $0$. Therefore, for large enough $m$, we have

$$\mathtt{ENT}(G) \approx \log Z = \log \left( \sum_{\boldsymbol{x} \in \mathsf{val}(\mathbf{X})} \sigma\left(w_0 + \sum_{i=1}^n w_i \cdot x_i\right) \right) = \log \left( \sum_{\boldsymbol{x} \in \mathsf{val}(\mathbf{X})} \sigma\left(w_0 + \sum_{i=1}^n w_i \cdot x_i\right) \right).$$

For each $S \subseteq [n]$, we define $\mathrm{weight}(S) := -\frac{m}{2} - mc + m(\sum_{i \in S} k_i)$. Then,

$$\exp(\mathtt{ENT}(G)) \approx \sum_{\boldsymbol{x} \in \mathsf{val}(\mathbf{X})} \sigma\left(-\frac{m}{2} - mc + m(\sum_{i \in [n]} k_i x_i)\right)$$

$$= \sum_{\boldsymbol{x} \in \mathsf{val}(\mathbf{X})} \sigma\left(-\frac{m}{2} - mc + m(\sum_{i : x_i = 1} k_i)\right)$$

$$= \sum_{S \subseteq [n]} \sigma(\mathrm{weight}(S))$$

$$= \frac{1}{2} \sum_{S \subseteq [n]} \left( \sigma(\mathrm{weight}(S)) + \sigma(\mathrm{weight}(\bar{S})) \right).$$

If $S$ is a solution to #NUMPAR, then

$$\sigma(\mathrm{weight}(S)) + \sigma(\mathrm{weight}(\bar{S})) = 2\sigma(-m/2).$$

Otervise, one of $\mathrm{weight}(S)$ and $\mathrm{weight}(\bar{S})$ is $\geq m/2$ and the other is $\leq -3m/2$, and hence

$$\sigma(m/2) \leq \sigma(\mathrm{weight}(S)) + \sigma(\mathrm{weight}(\bar{S})) \leq 1 + \sigma(-3m/2).$$

For a large enough $m$ such that $2\sigma(-m/2) < \epsilon$ and $1 - \sigma(m/2) < \epsilon$, we have

$$S \in P: \quad 0 \leq \sigma(\mathrm{weight}(S)) + \sigma(\mathrm{weight}(\bar{S})) \leq \epsilon,$$
$$S \notin P: \quad 1 - \epsilon \leq \sigma(\mathrm{weight}(S)) + \sigma(\mathrm{weight}(\bar{S})) \leq 1 + \epsilon.$$

Therefore, we have

$$\frac{2^n - |P|}{2}(1 - \epsilon) \leq \exp(\texttt{ENT}(G)) \leq \frac{|P|}{2}\epsilon + \frac{2^n - |P|}{2}(1 + \epsilon)$$

$$|P| \geq 2^n - \frac{2\exp(\texttt{ENT}(G))}{1 - \epsilon}$$

$$|P| \leq 2^n(1 + \epsilon) - 2\exp(\texttt{ENT}(G))$$

This gives a lower and upper bound for $|P|$. For small enough $\epsilon$ (governed by large enough $m$), the difference between the lower and upper bound is less than 1, and hence $|P|$ can be uniquely determined, which proves the theorem.

### A.5   Proof of Theorem 4

First note that according to Lem. 2, Eq. (2) is not a convex optimization problem. The key idea of Alg. 3 is to propose a set of *surrogate objective* functions, and maximize the objective function Eq. (2) by iteratively maximizing the surrogate objective. Concretely, we show the monotonic convergence property of Alg. 3 by checking the correctness of the following three statements:

• **Statement #1:** The surrogate objective is easy to maximize as it is a concave function w.r.t. the parameters.

• **Statement #2:** The surrogate objective is consistent with the original objective Eq. (2). That is, whenever a set of surrogate objectives are improved, the true objective is also improved.

• **Statement #3:** The surrogate objectives can always be improved unless the original objective Eq. (2) has zero first-order derivative.

• **Statement #4:** Solving Eq. (5) is equivalent to maximizing the surrogate objective.

Before verifying the statements, we first formally define the surrogate. Denote $\texttt{ENT}(p_n; \boldsymbol{\theta})$ as the entropy of the PC rooted at $n$ and with parameters $\boldsymbol{\theta}$; the top-down probability of $n$, denoted $P_{n_r}(n)$, is recursively defined as follows:

$$P_{n_r}(n) := \begin{cases} 1 & \text{if n is the root node } n_r, \\ \sum_{m \in \texttt{pa}(n)} P_{n_r}(m) & \text{if n is a sum unit,} \\ \sum_{m \in \texttt{pa}(n)} \theta_{m,n} \cdot P_{n_r}(m) & \text{if n is a product unit.} \end{cases} \quad (21)$$

Given a set of reference parameters $\boldsymbol{\theta}^{\text{ref}}$, we define the surrogate objective w.r.t. parameter $\theta_{n,c}$ as

$$\mathcal{L}_{\text{surr}}(\theta_{n,c}; \boldsymbol{\theta}^{\text{ref}}) := \underbrace{\frac{1}{N}\sum_{i=1}^{N} \log p(\boldsymbol{x}^{(i)}; \boldsymbol{\theta}^{\text{ref}}\backslash\{\theta_{n,c}^{\text{ref}}\}, \theta_{n,c})}_{\text{Term 1}}$$

$$+ \underbrace{\tau \cdot P_{n_r}(n; \boldsymbol{\theta}^{\text{ref}}) \cdot \left(-\theta_{n,c}\log\theta_{n,c} + \theta_{n,c}\cdot\texttt{ENT}(p_c; \boldsymbol{\theta}^{\text{ref}})\right)}_{\text{Term 2}}. \quad (22)$$

Given parameters $\boldsymbol{\theta}^{\text{old}}$, we now describe an update procedure to obtain a set of new parameters $\boldsymbol{\theta}^{\text{new}}$.

**Parameter update procedure**   We start with an empty set of parameters $\boldsymbol{\theta}^{\text{update}} := \boldsymbol{\theta}^{\text{old}}$ and iteratively update its entries with updated parameters $\theta_{n,c}^{\text{new}}$. For every sum unit $n$ traversed in pre-order, we update the parameters $\{\theta_{n,c} \mid c \in \texttt{in}(n)\}$ by maximizing the sum of surrogate objectives:

$$\sum_{c \in \texttt{in}(n)} \mathcal{L}_{\text{surr}}(\theta_{n,c}; \boldsymbol{\theta}^{\text{update}}). \quad (23)$$

After solving the above equation, the updated parameters $\{\theta_{n,c} \mid c \in \texttt{in}(n)\}$ replace the corresponding original parameters in $\boldsymbol{\theta}^{\text{update}}$. As we will proceed to show in statement #4, maximizing Eq. (23) is done in Lines 7 to 7 in Alg. 3.

Given the formal definition of the surrogate objective and the corresponding update process, we re-state the three statements and prove their validity in the following.

• **Statement #1:** The surrogate objective Eq. (23) is concave w.r.t. parameters $\{\theta_{n,c} \mid c \in \texttt{in}(n)\}$.

*Proof.* This statement can be proved by showing that $\forall (n, c), \forall \boldsymbol{\theta}, \mathcal{L}_{\text{surr}}(\theta_{n,c}; \boldsymbol{\theta})$ is concave. Specifically, according to Lem. 3, the first term of Eq. (22) is concave; the second term of Eq. (22) is concave since (i) $-x \log x$ is concave w.r.t. $x$, and (ii) $P_{n_r}(n; \boldsymbol{\theta}^{\text{ref}})$ and $\text{ENT}(p_c; \boldsymbol{\theta}^{\text{ref}})$ are independent of $\{\theta_{n,c'} \mid c' \in \text{in}(n)\}$. $\qquad\square$

• **Statement #2:** For any sum unit $n$ and any parameters $\boldsymbol{\theta}$, if we update $n$'s parameters (i.e., $\{\theta_{n,c} \mid c \in \text{in}(n)\}$) by maximizing Eq. (23), the true objective Eq. (2) will also improve.

*Proof.* Consider updating the parameters correspond to sum unit $n$ (i.e., $\{\theta_{n,c} \mid c \in \text{in}(n)\}$) by maximizing Eq. (23). We can re-arrange the entropy $\text{ENT}(p_{n_r})$ as follows:

$$
\begin{aligned}
\text{ENT}(p_{n_r}) &\overset{(a)}{=} - \sum_{(n',c') \in \text{edges}(p_{n_r})} P_{n_r}(n') \cdot \theta_{n',c'} \log \theta_{n',c'} \\
&= - \sum_{(n',c') \in \text{edges}(p_n)} P_{n_r}(n') \cdot \theta_{n',c'} \log \theta_{n',c'} + \text{const} \\
&= - \sum_{(n',c') \in \text{edges}(p_n)} \left( \sum_{m \in \text{pa}(n)} P_{n_r}(m) \right) \cdot P_n(n') \cdot \theta_{n',c'} \log \theta_{n',c'} + \text{const} \\
&= - \sum_{(n',c') \in \text{edges}(p_n)} P_{n_r}(n) \cdot P_n(n') \cdot \theta_{n',c'} \log \theta_{n',c'} + \text{const} \\
&= P_{n_r}(n) \cdot \text{ENT}(p_n) + \text{const} \\
&\overset{(b)}{=} P_{n_r}(n) \cdot \sum_{c \in \text{in}(n)} \left( -\theta_{n,c} \log \theta_{n,c} + \theta_{n,c} \cdot \text{ENT}(p_c) \right) + \text{const},
\end{aligned}
$$

where const denotes terms that do not depend on $\{\theta_{n,c'} \mid c' \in \text{in}(n)\}$; $(a)$ and $(b)$ directly apply Lem. 4 and Lem. 1, respectively.

Thus, the true objective Eq. (2) can be written as follows:

$$
\begin{aligned}
&\frac{1}{N} \sum_{i=1}^{N} \log p(\boldsymbol{x}^{(i)}) + \tau \cdot \text{ENT}(p) \\
={} &\frac{1}{N} \sum_{i=1}^{N} \log p(\boldsymbol{x}^{(i)}) + \tau \cdot P_{n_r}(n) \cdot \sum_{c \in \text{in}(n)} \left( -\theta_{n,c} \log \theta_{n,c} + \theta_{n,c} \cdot \text{ENT}(p_c) \right) + \text{const} \qquad (24)
\end{aligned}
$$

Compare Eq. (24) and Eq. (22), we can see that they only differs in some constant terms. Therefore, maximizing Eq. (22) w.r.t. $\{\theta_{n,c'} \mid c' \in \text{in}(n)\}$ will lead to an increase in the true objective Eq. (2). $\qquad\square$

• **Statement #3:** The surrogate objectives can always be improved unless the original objective Eq. (2) has zero first-order derivative.

*Proof.* Recall from Eq. (24) that for any sum unit $n$, the true objective Eq. (2) can be written as the sum of Eq. (22) and terms that are independent with the parameters of $n$ (i.e., $\{\theta_{n,c'} \mid c' \in \text{in}(n)\}$). Therefore, the true objective can always be improved by maximizing the surrogate objective Eq. (23) as long as the true objective has non-zero first-order derivative w.r.t. the parameters. $\qquad\square$

• **Statement #4:** Solving Eq. (5) is equivalent to maximizing the surrogate objective.

*Proof.* We want to maximize the surrogate objective given the assumption that the parameters w.r.t. a sum unit sum up to 1:

$$
\underset{\theta_{n,c}}{\text{maximize}} \, \mathcal{L}_{\text{surr}}(\theta_{n,c}; \boldsymbol{\theta}^{\text{ref}}), \text{ such that } \sum_{c \in \text{in}(n)} \theta_{n,c} = 1. \qquad (25)
$$

| **Algorithm 5** Forward pass (expected flows) | **Algorithm 6** Backward pass (expected flows) |
|---|---|
| 1: **Input:** A non-deterministic PC $p$; sample $\boldsymbol{x}$ | 1: **Input:** A non-deterministic PC $p$; $\forall n, \texttt{value}[n]$ |
| 2: **Output:** $\texttt{value}[n] := (\boldsymbol{x} \in \mathsf{supp}(n))$ for each unit $n$ | 2: **Output:** $\texttt{eflow}[n,c] := \mathbb{E}_{\boldsymbol{z} \in p_c(\cdot \mid \boldsymbol{x}; \boldsymbol{\theta})}((\boldsymbol{x}, \boldsymbol{z}) \in$ |
| 3: **foreach** $n$ traversed in postorder **do** | $(\gamma_n \cap \gamma_c))$ for each pair $(n, c)$, where $n$ is a sum |
| 4: $\quad$ **if** $n$ **isa** input unit **then** $\texttt{value}[n] \leftarrow f_n(\boldsymbol{x})$ | unit and $c \in \mathsf{in}(n)$ |
| 5: $\quad$ **elif** $n$ **isa** product unit **then** | 3: $\forall n, \texttt{context}[n] \leftarrow 0; \texttt{context}[n_r] \leftarrow \texttt{value}[n_r]$ |
| 6: $\quad\quad$ $\texttt{value}[n] \leftarrow \prod_{c \in \mathsf{in}(n)} \texttt{value}[c]$ | 4: **foreach** sum unit $n$ traversed in preorder **do** |
| 7: $\quad$ **else** //$n$ is a sum unit | 5: $\quad$ **foreach** $m \in \mathsf{pa}(n)$ **do** (denote $g \leftarrow \mathsf{pa}(m)$) |
| 8: $\quad\quad$ $\texttt{value}[n] \leftarrow \sum_{c \in \mathsf{in}(n)} \theta_{n,c} \cdot \texttt{value}[c]$ | 6: $\quad\quad$ $\texttt{f} \leftarrow \frac{\texttt{value}[m]}{\texttt{value}[g]} \cdot \texttt{context}[g] \cdot \theta_{g,m}$ |
| | 7: $\quad\quad$ $\texttt{context}[n] \mathrel{+}= \texttt{f}; \quad \texttt{flow}[g,m] = \texttt{f}$ |

Since the surrogate objective $\mathcal{L}_{\text{surr}}(\theta_{n,c}; \boldsymbol{\theta}^{\text{ref}})$ is concave, maximizing the surrogate objective is equivalent to finding its stationary point. Specifically, we solve Eq. (25) with the Lagrange multiplier method (variable $\lambda$ corresponds to the constraint):

$$\underset{\theta_{n,c}}{\text{maximize}} \, \underset{\lambda}{\text{minimize}} \, \mathcal{L}_{\text{surr}}(\theta_{n,c}; \boldsymbol{\theta}^{\text{ref}}) - \lambda(1 - \sum_{c \in \mathsf{in}(n)} \theta_{n,c})$$

Its KKT conditions can be written as:

$$\begin{cases} \frac{\mathrm{F}_{n,c_i}(\mathcal{D})}{|\mathcal{D}| \cdot \theta_{n,c_i}} - \tau \cdot P_{n_r}(n; \boldsymbol{\theta}^{\text{ref}})(\log \theta_{n,c_i} + 1 + \mathtt{ENT}(p_{c_i}; \boldsymbol{\theta}^{\text{ref}})) + \lambda = 0 & (\forall 1 \leq i \leq |\mathsf{in}(n)|), \\ \sum_{c \in \mathsf{in}(n)} \theta_{n,c} = 1. \end{cases}$$

It is easy to verify that the above equation is equivalent to Eq. (5) by substituting the definitions in Lines 7-8 in Alg. 3.

$\square$

Therefore, by following the parameter update procedure, we can always make progress since the surrogate objective is concave (statement #1) and the true objective improves as long as the surrogate objective increases (statement #2). Finally, the learning procedure will not terminate unless a local maximum is achieved (statement #3).

### A.6 Correctness of Algorithms 1 and 2

The correctness of Alg. 1 and 2 can be justified directly by the proof of Thm. 3. Specifically, since with $\beta = 1$, the softened dataset $\mathcal{D}_\beta$ is equivalent to $\mathcal{D}$, we can use the proof in Appendix A.1 and set $\beta = 1$ (the proof holds for any $\beta \in (0.5, 1]$).

### A.7 Proof of Proposition 1

The first statement (i.e., Eq. (2)) could be non-concave) is proved in Lem. 2. The second statement (i.e., Eq. (2) could have multiple local maximas) is proved in Lem. 5.

## B Method or Experiment Details

### B.1 Soften non-boolean datasets

As a direct extension of softening boolean datasets, datasets with categorical variables can be similarly softened. Suppose $X$ is a categorical variable with $k$ categories. For an assignment $x = j$, we can soften it as follows

$$\begin{cases} P(x = i) = \frac{1-\beta}{k} & (i \neq j), \\ P(x = j) = \beta. \end{cases}$$

To compute the flow $F_{n,c}(\mathcal{D}_\beta)$ w.r.t. a softened categorical dataset, we can again adopt Alg. 1 and 2 by choosing

$$f_n(\boldsymbol{x}) = \beta \cdot \mathbb{1}[\boldsymbol{x} \in \mathsf{supp}(n)] + \frac{1 - \beta}{k} \cdot \mathbb{1}[\boldsymbol{x} \notin \mathsf{supp}(n)].$$

## B.2 Solving Equation 5

Denote $\gamma_{c_i} := \texttt{entropy}[c_i]$, our goal is to solve the following set of equations:

$$\begin{cases} d_i e^{-\varphi_{n,c_i}} - b \cdot \varphi_{n,c_i} + b \cdot \gamma_{c_i} = y & (\forall i \in \{1, \ldots, |\mathsf{in}(n)|\}), \\ \sum_{i=1}^{|\mathsf{in}(n)|} e^{\varphi_{n,c_i}} = 1. \end{cases}$$

We break down the problem by iteratively solve for $\{\varphi_{n,c_i}\}_{i=1}^{|\mathsf{in}(n)|}$ and $y$, respectively.

• Solve for $y$. Given variables $\{\varphi_{n,c_i}\}_{i=1}^{|\mathsf{in}(n)|}$, we update $y$ as

$$y = \frac{1}{|\mathsf{in}(n)|} \sum_{i=1}^{|\mathsf{in}(n)|} d_i e^{-\varphi_{n,c_i}} - b \cdot \varphi_{n,c_i} + b \cdot \gamma_{c_i}.$$

• Solve for $\{\varphi_{n,c_i}\}_{i=1}^{|\mathsf{in}(n)|}$. Given $y$, we first update each $\varphi_{n,c_i}$ individually by solving the equation

$$d_i e^{-\varphi_{n,c_i}} - b \cdot \varphi_{n,c_i} + b \cdot \gamma_{c_i} = y.$$

Specifically, this is done by iterative Newton method update:

$$\varphi_{n,c_i} \mathrel{+}= \frac{\frac{d_i}{\varphi_{n,c_i}} + b \cdot (\gamma_{c_i} - \varphi_{n,c_i}) + y}{\frac{d_i}{\varphi_{n,c_i}} + b}$$

After one Newton method update step for every parameter in $\{\varphi_{n,c_i}\}_{i=1}^{|\mathsf{in}(n)|}$, we enforce the constraint $\sum_{i=1}^{|\mathsf{in}(n)|} e^{\varphi_{n,c_i}} = 1$ by

$$\varphi_{n,c_i} \mathrel{-}= \log \Big( \sum_{i=1}^{|\mathsf{in}(n)|} e^{\varphi_{n,c_i}} \Big).$$

## B.3 Details of the Experiments on Deterministic PCs

**PC structures** For each dataset, we adopt 16 PCs by running Strudel [17] for $\{1000, 1200, 1400, \ldots, 4000\}$ iterations except for the dataset "dna", which we ran Strudel for $\{50, 100, 150, \ldots, 800\}$ iterations since the learning algorithm takes significantly longer for this dataset.

**Hyperparameters** We always perform hyperparameter search using the validation set, and report the final performance on the test set. Whenever we use data softening or entropy regularization, we also add pseudocount $\alpha = 1$ since it yields better performance.

**Server specifications** All our experiments were run on a server with 72 CPUs, 512G Memory, and 2 TITAN RTX GPUs.

**Detailed results** See Table 3 for extended numerical results.

## B.4 Details of the Experiments on Non-Deterministic PCs

**The HCLT structure** For the experiments on the twenty datasets, we set the hidden size of the HCLT structure as 12, i.e., every latent variable $Z$ is a categorical variable with 12 categories. Additionally, following [17, 16], we learn a mixture of 4 HCLTs to achieve better performance. For the protein sequence dataset, we adopted a mixture of 2 HCLTs with hidden size 32.

**Hyperparameters** Due to the complexity of running EM iteratively, we were not able to perform a grid-search for hyperparameters since that would take too long. In our experiments, we tried the following sets of hyperparameters (for $\alpha$, $\beta$, and $\tau$): $(0.1, 1.0, 0.0)$, $(0.2, 1.0, 0.0)$, $(0.4, 1.0, 0.0)$, $(0.1, 0.998, 0.0)$, $(0.1, 1.0, 0.001)$, and $(0.1, 0.998, 0.001)$. Among these hyperparameter choices, $(0.1, 0.998, 0.001)$ achieved the best validation LL in most datasets, and thus we reported this set of results. Therefore, for non-deterministic PCs, it is also beneficial to combine both proposed regularization techniques.

Table 2: Full results on the 20 density estimation benchmarks. As an extension of Table 1, we report the average test-set log-likelihood of all baselines: Strudel [17], LearnPSDD [16], EinSumNet [13], LearnSPN [18], ID-SPN [47], and RAT-SPN [48].

| Dataset | HCLT | EiNet | LearnSPN | ID-SPN | RAT-SPN | Strudel | LearnPSDD |
|---|---|---|---|---|---|---|---|
| accidents | -26.78 | -35.59 | -40.50 | -26.98 | -35.48 | -29.46 | -28.29 |
| ad | -16.04 | -26.27 | -19.73 | -19.00 | -48.47 | -16.52 | -20.13 |
| baudio | -39.77 | -39.87 | -40.53 | -39.79 | -39.95 | -42.26 | -41.51 |
| bbc | -250.07 | -248.33 | -250.68 | -248.93 | -252.13 | -258.96 | -260.24 |
| bnetflix | -56.28 | -56.54 | -57.32 | -56.36 | -56.85 | -58.68 | -58.53 |
| book | -33.84 | -34.73 | -35.88 | -34.14 | -34.68 | -35.77 | -36.06 |
| c20ng | -151.92 | -153.93 | -155.92 | -151.47 | -152.06 | -160.77 | -160.43 |
| cr52 | -84.67 | -87.36 | -85.06 | -83.35 | -87.36 | -92.38 | -93.30 |
| cwebkb | -153.18 | -157.28 | -158.20 | -151.84 | -157.53 | -160.50 | -161.42 |
| dna | -79.33 | -96.08 | -82.52 | -81.21 | -97.23 | -87.10 | -83.02 |
| jester | -52.45 | -52.56 | -75.98 | -52.86 | -52.97 | -55.30 | -54.63 |
| kdd | -2.18 | -2.18 | -2.18 | -2.13 | -2.12 | -2.17 | -2.17 |
| kosarek | -10.66 | -11.02 | -10.98 | -10.60 | -10.88 | -10.98 | -10.99 |
| msnbc | -6.05 | -6.11 | -6.11 | -6.04 | -6.03 | -6.05 | -6.04 |
| msweb | -9.90 | -10.02 | -10.25 | -9.73 | -10.11 | -10.19 | -9.93 |
| nltcs | -6.00 | -6.01 | -6.11 | -6.02 | -6.01 | -6.06 | -6.03 |
| plants | -14.31 | -13.67 | -12.97 | -12.54 | -13.43 | -13.72 | -13.49 |
| pumbs* | -23.32 | -31.95 | -24.78 | -22.40 | -32.53 | -25.28 | -25.40 |
| tmovie | -50.69 | -51.70 | -52.48 | -51.51 | -53.63 | -59.47 | -55.41 |
| tretail | -10.84 | -10.91 | -11.04 | -10.85 | -10.91 | -10.90 | -10.92 |

Table 3: Results comparing different regularization approaches using the 20 density estimation benchmarks. This table contains the part of the results summarized in Fig. 5. Specifically, we report performance of the PC generated by running Strudel [17] for 4,000 steps, except for dna, where we ran the learner for 1,000 steps.

| Dataset | Laplace smoothing | Data softening | Entropy reg. | Data softening + Entropy reg. |
|---|---|---|---|---|
| accidents | -29.37 | -29.37 | -29.39 | -29.37 |
| ad | -16.39 | -16.39 | -16.55 | -16.39 |
| baudio | -42.89 | -42.75 | -42.78 | -42.59 |
| bbc | -258.82 | -258.64 | -258.71 | -258.35 |
| bnetflix | -59.51 | -59.34 | -59.19 | -59.07 |
| book | -36.93 | -36.81 | -37.05 | -36.69 |
| c20ng | -160.84 | -160.80 | -160.81 | -160.73 |
| cr52 | -91.97 | -91.91 | -91.99 | -91.86 |
| cwebkb | -159.93 | -159.78 | -159.97 | -159.67 |
| dna | -95.63 | -94.90 | -95.24 | -94.87 |
| jester | -56.19 | -55.95 | -55.83 | -55.62 |
| kdd | -2.19 | -2.18 | -2.19 | -2.17 |
| kosarek | -11.03 | -11.00 | -11.04 | -10.97 |
| msnbc | -6.04 | -6.04 | -6.04 | -6.04 |
| msweb | -10.11 | -10.08 | -10.12 | -10.06 |
| nltcs | -6.18 | -6.10 | -6.17 | -6.09 |
| plants | -13.56 | -13.42 | -13.56 | -13.41 |
| pumbs* | -25.66 | -25.66 | -25.69 | -25.68 |
| tmovie | -59.56 | -59.44 | -59.53 | -59.35 |
| tretail | -11.34 | -11.30 | -11.35 | -11.27 |

**Detailed results**   As an extension of Table 1, Table 2 provides the average test set log-likelihood for all adopted baselines.

**Hyperparameters of RAT-SPN**   We took the RAT-SPN results on the twenty density estimation benchmarks from the original paper. Therefore, the hyperparameter settings for RAT-SPN are the same as reported in the original paper: cross-validate the split-depth $D \in \{1, 2, 3, 4\}$ and the number of sum-weights $W_s \in \{1e3, 1e4, 1e5\}$, and used Eq. (1) in [48] to select R, S, and I. Following the original paper, dropout is not used for training the generative models.