# OpenReview forum: "Tractable Regularization of Probabilistic Circuits"
_NeurIPS.cc/2021/Conference — NeurIPS 2021 Spotlight_

### Official Review · Reviewer_EufT · 2021-07-15

**Rating:** 7
**Confidence:** 4

**Summary:**

In the context of Probabilistic Circuits (PCs), recent developments of large-scale models point to the increasing need of regularization.
Having large models with millions of parameters is inherently a challenge, but more so for models with structural restrictions that favor tractability.
In this paper, the authors present regularization techniques for probabilistic circuits including non-deterministic models.
The regularization is based on data softening and on regularization via entropy, and the authors show that this is possible for PCs while intractable for other models.
And demonstrate empirically that the approach is beneficial.

**Ethical Concerns:**

No ethical concerns.

**Limitations And Societal Impact:**

The authors have not mentioned societal impact, but I believe there are no conflicts or points to raise.

**Main Review:**

The authors present a regularization approach based on a modified loss that includes an entropy component. Furthermore, updates based on a softened dataset are computed in time linear w.r.t. to the model size and the training dataset size.

The paper is well-written and technically sound, and very relevant and significant for the tractable probabilistic models community.
The authors give a proper introduction to the models and the different characterizations based on determinism. Moreover, they also show how this regularization approach would not be tractable for other ML approaches. In short, this approach is uniquely tailored to the capabilities of probabilistic circuits.

There are minor clarifications that I believe could help readers better understand the paper:
- In lines 48/49 (and later), you mention that MLE parameters can be obtained in linear time wrt to sizes. But you present an iterative approach in Eq. 4. This would introduce another factor in the time needed for the MLE estimation. I believe the distinction is based on determinism, and this could be mentioned earlier (e.g., in lines 48/49).
- In line 141, maybe you want to add the word “exact”, as you can compute approximates for the non-deterministic variant.

A more important question is the comparison to other regularization approaches, and the most significant here is dropout as introduced in RAT-SPNs. In your paper you mention that you compared to this approach (RAT), but you do not mention whether dropout was enabled nor what hyper-parameters were used for that.
A comparison here would help the reader better understand the impact of your approach, even if they are orthogonal.

**Time Spent Reviewing:**

3

---

> ### Author Response · Authors · 2021-08-10
> **Response to Reviewer EufT**
>
> We thank the reviewer for their time spent reviewing our work and for evaluating it to be well-written and technically sound.
>
>  - Clarification of Lines 48-49:
>
> Thank you for spotting this presentation issue. As formally stated in Thm. 3, the linear run-time of data softening is achievable when the PC is deterministic, and for non-deterministic we have to account for the extra complexity brought by EM (Sec. 4.3). We will revise the presentation in Lines 48-49 to better demonstrate this fact.
>
>  - Clarification of Line 141:
>
> Thanks for your suggestion. We will add the word “exact” to highlight the difference with approximate MAP algorithms.
>
>  - Hyperparameters of RAT-SPN
>
> We took the RAT-SPN results on the twenty density estimation benchmarks from the original paper. Therefore, the hyperparameter settings for RAT-SPN are the same as reported in the original paper: cross-validate the split-depth D \in {1,2,3,4} and the number of sum-weights W_s \in {1e3,1e4,1e5}, and used Eq. (1) in [1] to select R, S, and I. Following the original paper [1], dropout is not used for training the generative models. We will add the above information in the revised paper.
>
> Also, we talked to one of the authors of the RAT-SPN paper, and they told us they did not observe much gain on LL by using dropout for the generative PCs. Dropout helped mostly in the discriminative case (e.g., classification).
>
> [1] Peharz, Robert, et al. "Random sum-product networks: A simple and effective approach to probabilistic deep learning." Uncertainty in Artificial Intelligence. PMLR, 2020.

---

### Official Review · Reviewer_F3ZK · 2021-07-17

**Rating:** 7
**Confidence:** 3

**Summary:**

This paper tackles a common issue with probabilistic circuits (PCs) in practice: overfitting. The work suggests two novel methods for principled regularization. The first method, called data softening, injects noise into the dataset. The second method, called entropy regularization, directly regularizes the entropy of the distribution encoded by the PC. Both methods are global and model-agnostic, meaning they can be applied to any arbitrary distribution, regardless of the model.

**Limitations And Societal Impact:**

This paper does not explicitly discuss the limitations of the method's proposed. Given the theoretical results presented here, a formal analysis of possible assumptions or requirements for the method's application would be welcome.

**Main Review:**

This work is original in the PC context. Both regularization techniques are not novel ideas on their own. Indeed, data softening is related to data augmentation, a procedure wildly used in machine learning. Similarly, entropy as an overfitting measurement is also not innovative. However, bringing these intuitive methods to PC regularization is a clever move. PCs allow for (mostly) tractable inference, and the two regularization techniques benefit from that tractability. Therefore, the methods proposed in the manuscript are new and of relevance.

This paper has a good technical quality. The main ideas in Section 2 are introduced with proper formalism. Similarly, the work's main theoretical contributions in Section 4 are formally demonstrated and proved. Specifically, Theorem 4 shows the monotonical convergence of Algorithm 3, a result of practical interest. All derived results solidify their principled proposed methods.

The manuscript is well-written and clear. I also enjoyed the presentation, which favours the logical flow of explanation. For example, empirical results are given right after the method's description, as in Sections 4.2 and 4.3

The experiment results demonstrate the work's relevance. While reaching state-of-the-art in some datasets, the experiment results section can be strengthened with tasks evaluation. At times, log-likelihood will not reflect performance in specific probabilistic tasks, such as classification or regression. Thus, it would be interesting to see the influence of the proposed regularization techniques in such tasks.

**Time Spent Reviewing:**

1

---

> ### Author Response · Authors · 2021-08-10
> **Response to Reviewer F3ZK**
>
> We thank the reviewer for their time and for evaluating our paper as having good technical quality.
>
>  - Experimenting with other tasks:
>
> Thank you for the suggestion. It would be nice to see how the proposed regularization methods work in other types of tasks such as classification and regression, and we will spend more time investigating them in future work. In this paper, we used classic density estimation benchmarks to show that the proposed techniques can be used to learn distributions that generalize better, which is an important property of machine learning models.
>
>  - Discussion on the method’s application:
>
> Our theoretical results in Section 4 provide a relatively precise characterization of the performance of both proposed methods when applied to deterministic PCs. However, due to the extra complexity introduced by EM, it is hard to theoretically analyze the performance of proposed methods on non-deterministic PCs. Therefore, we use empirical evaluations to supplement the theoretical results: from the experiment results in Fig. 6 and Table 1, we empirically show that both data softening and entropy regularization do benefit the learning of non-deterministic PCs consistently.

---

### Official Review · Reviewer_DV79 · 2021-07-18

**Rating:** 7
**Confidence:** 5

**Summary:**

The paper deal with the problem of regularization of Probabilistic Circuits (PCs).

**Limitations And Societal Impact:**

Sec. 4.2 discusses the limitation489of Thm. 4; Sec. 4.3 discusses the limitation of applying entropy regularization to490non-deterministic PCs

**Main Review:**

Two approaches have been proposed, one based on softening the training samples and the other one based on controlling the entropy of the learned distribution. They show that both the proposed approaches improve the test set likelihood over standard density estimation datasets.

Some theorems are presented in order to prove the tractability of the learning algorithms.

Table 1 is not referenced in the paper.

As regards the Laplace smoothing, the problem of uniform prior has been addressed in [1], where the authors introduced a Bayesian Parameter Smoothing for deterministic PCs.

It is not clear why a simple CLT has been used for non-deterministic PCs instead of learning a more complex one.

The experimental evaluation proves the validity of the proposed approaches.

Concluding, the paper is well written, the idea is novel for PCs, and the results, both theoretical and empirical, are good.

[1] Learning accurate cutset networks by exploiting decomposability, 2015


**Time Spent Reviewing:**

3

---

> ### Author Response · Authors · 2021-08-10
> **Response to Reviewer DV79**
>
> We thank the reviewer for considering our paper to be novel and containing good empirical and theoretical results.
>
>  - Reference to Table 1.
>
> We thank the reviewer for spotting this missing reference. In the revised paper we will add a reference to Table 1 in Lines 338-341, where the results in the table are discussed.
>
>  - The uniform prior problem.
>
> Thank you for pointing out this relevant work. In [1], the Bayesian Parameter Smoothing method works well in learning the parameters of CLTs. With some generalization and extensions, the idea of Bayesian parameter smoothing can also be applied to PCs. We will mention in the revised paper that Bayesian Parameter Smoothing is a solution to the imbalanced problem caused by Laplace smoothing and discuss the connections between Bayesian Parameter Smoothing and proposed regularization techniques.
>
>  - Why not use a learned non-deterministic PC.
>
> There might be a misunderstanding on the complexity of HCLTs. Although CLTs learned from data are used to construct HCLTs, they are drastically different in terms of expressiveness and structural properties. Specifically, CLTs can be compiled into deterministic PCs, while HCLTs are non-deterministic. HCLTs are essentially complex hidden-variable models that are much more expressive than CLTs. HCLTs can be thought of as structured hidden-variable models where the CLT structure is used to put highly correlated variables closer.
>
> To confirm the complexity of HCLTs, we note that the sizes of the adopted HCLTs are on par with the baselines. For example, for the dna dataset (in the 20-datasets), the HCLT has ~100,000 edges. This is at the same scale as the size of the baseline PCs. In the revised paper, we will add a table to compare the PC sizes.
>
> Therefore, the adopted HCLTs are large and expressive enough to demonstrate the effect of the proposed regularization approaches. As an example, in Fig. 6, the proposed regularization methods significantly improved the average test LL on a protein dataset.

---

### Official Review · Reviewer_HNkH · 2021-07-22

**Rating:** 7
**Confidence:** 3

**Summary:**

This is a paper about regularizing probabilistic circuits (PCs), a rich class of tractable probabilistic models. The authors propose two new methods: data softening and entropy regularization. Data softening aims to put a non-zero mass on each possible assignment with a higher emphasis on observed assignments in the given data. Training the PC on this soften data implicitly regularizes its parameters. Entropy regularization aims to avoid learning low-entropy distributions which are prone to overfitting. To achieve this the authors introduce an entropy regularizer into the log-likelihood objective function. Both techniques are computationally intractable to implement even for simple machine learning models. The authors leverage the tractability and structural properties of PCs and propose how the techniques can be practically applied to learn regularized parameters. On a number of benchmark datasets for density estimation, the proposed methods achieved improved generalization accuracy compared to other widely used regularization techniques.

**Limitations And Societal Impact:**

yes

**Main Review:**

Laplace smoothing has been the most widely used technique for regularizing the parameters of both deterministic and non-deterministic PCs. The authors give an example of how this simple technique can fail to produce reasonable non-uniform priors for sum units in PCs that are imbalanced (have drastically different support sizes). The authors make two contributions to address this issue. The first contribution is softening the data which implicitly regularizes the parameters of the PC by adding noise in closed form. The authors have theoretically analyzed the non-trivial challenge of training machine learning models using such softened data and proved that PCs are an exception to this: One can train a PC using softened data (exponentially many) in time that is linear in its size as well as the original data. The second contribution is proposing an entropy regularization objective function that essentially adds the entropy of the model (with a hyperparameter) to the log-likelihood objective function to be maximized, essentially discouraging learning models with lower entropy. This was also proven to be a computationally hard task when the model is not a PC. The authors have presented an EM-style algorithm to find a stationary point of the non-concave objective function. Both of the proposed techniques were applied to two PCs types: deterministic and non-deterministic. Experimental results showed that both approaches significantly improved the test set log-likelihood scores on several problems compared to Laplace smoothing.

Overall, the paper makes useful practical contributions for regularizing PCs. The motivation was well-expressed, challenges of the proposed approach were described and solutions for both types of PCs were presented. Empirical results were promising. I am writing some comments/suggestions that might improve the readability of the paper.

Comments:
1. Figure 5 shows summary results of regularizing several PC structures using both data softening and entropy regularization. Since it’s a summary result its hard to get a picture of how these methods worked for higher dimensional problems and what were there structural complexities. I would suggest to present a table similar to Table 1.
2. Proof of theorem 3 was quite involved. It would help if a high-level sketch is given first and then details are presented.
3. Table 1 only reports one set of hyperparameters. It was unclear whether any tuning was performed here. Also, the \beta hyperparameter is shown to be very small (<0.5) compared to the ones used for deterministic PCs. Were the two techniques combined for HCLTs?

**Time Spent Reviewing:**

14

---

> ### Author Response · Authors · 2021-08-10
> **Response to Reviewer HNkH**
>
> Thank you very much for your detailed review and for praising its empirical and theoretical contributions. Below are the responses to your three detailed comments.
>
>  - Response to comment #1 (detailed results in Fig. 5):
>
> Thank you for the helpful suggestion.  In addition to Fig. 5, we agree that a table containing test set LLs for all 20 datasets w.r.t. different regularization methods would provide readers a better overview of how well both proposed techniques work. In the next version of the paper, a table with the full results will be added.
>
> - Response to comment #2 (proof of Thm. 3 is quite involved):
>
> The high-level idea of the proof of Thm. 3 is by separately showing the correctness of the forward pass (Alg. 1) and the backward pass (Alg. 2). Specifically, for a “softened” sample x, we aim to show that (i) in the forward pass, the value of x w.r.t. any PC node n corresponds to the likelihood of x (note that since x can be represented as a weighted sum of exponentially many “hard” samples, the target likelihood is also the weighted sum of the respective likelihoods), and (ii) in the backward pass, the flow of x w.r.t. any PC node corresponds to the weighted sum of the flows of the “hard” samples “contained” in x. Both claims are proved by induction: for the forward pass, we first show that the base cases (leaf nodes) satisfy the claim, then by assuming all children of a node satisfy the claim, we prove the inductive case of sum and product nodes; for the backward pass, induction is also applied, though in the preorder (parents before children).
>
> In the next version, we will add the above high-level sketch to the proof, and also provide more intuitive explanations of the target “values” (Eq. (6)) and “flows” (Eq. (8)) by showing why they are the “correct” target.
>
>  - Response to comment #3 (hyperparameter tuning for non-deterministic PC):
>
> There was a typo for the hyperparameter \beta in Table 1. The correct number should be 1-0.002 = 0.998. Thank you so much for spotting this, and it will be fixed in the next version.
>
> In the non-deterministic PC experiments, due to the complexity of running EM iteratively, we were not able to perform a grid-search for hyperparameters since that would take too long. In our experiments, we tried the following sets of hyperparameters (for \alpha, \beta, and \tau): (0.1, 1.0, 0.0), (0.2, 1.0, 0.0), (0.4, 1.0, 0.0), (0.1, 0.998, 0.0), (0.1, 1.0, 0.001), and (0.1, 0.998, 0.001). Among these hyperparameter choices, (0.1, 0.998, 0.001) achieved the best validation LL in most datasets, and thus we reported this set of results. Therefore, for non-deterministic PCs, it is also beneficial to combine both proposed regularization techniques. We will add the above discussion and hyperparameter specifications in the revised paper.

---

### Decision · Program_Chairs · 2021-09-27

**Decision:**

Accept (Spotlight)

**Comment:**

Thanks for submitting your work to NeurIPS. Getting probabilistic circuits (PCs) to scale well is important Here, the paper makes two important contributions to regularization of PCs, namely, data softening and entropy regularization. While both methods are infeasible for many machine learning models, the paper proves that  they can be efficiently implemented for PCs. All reviewers agree that this is a very solid and important contribution. And I fully agree with this sentiment.